METHODS

# Estimating receptive fields of simple and complex cells in early visual cortex: A convolutional neural network model with parameterized rectification

**Philippe Nguyen**[1¤a‡], **Jinani Sooriyaarachchi**[2‡], **Qianyu Huang**[3¤b], **Curtis L. Baker, Jr.**[4]*

**1** Department of Biomedical Engineering, McGill University, Montreal, Quebec, Canada, **2** Department of Physiology, McGill University, Montreal, Quebec, Canada, **3** Department of Biology, McGill University, Montreal, Quebec, Canada, **4** Department of Ophthalmology and Visual Sciences, McGill University, Montreal, Quebec, Canada

¤a Current address: Resson Aerospace | AgTech, New Brunswick, Canada
¤b Current address: SSW Consulting, Hangzhou, Zhejiang Province, China
‡ These authors share first authorship on this work.
* curtis.baker@mcgill.ca

**Data Availability Statement:** Data for figures, as well as stimuli and response data for an example neuron, may be found at: https://github.com/

## Abstract

Neurons in the primary visual cortex respond selectively to simple features of visual stimuli, such as orientation and spatial frequency. Simple cells, which have phase-sensitive responses, can be modeled by a single receptive field filter in a linear-nonlinear model. However, it is challenging to analyze phase-invariant complex cells, which require more elaborate models having a combination of nonlinear subunits. Estimating parameters of these models is made additionally more difficult by cortical neurons' trial-to-trial response variability. We develop a simple convolutional neural network method to estimate receptive field models for both simple and complex visual cortex cells from their responses to natural images. The model consists of a spatiotemporal filter, a parameterized rectifier unit (PReLU), and a two-dimensional Gaussian "map" of the receptive field envelope. A single model parameter determines the simple vs. complex nature of the receptive field, capturing complex cell responses as a summation of homogeneous subunits, and collapsing to a linear-nonlinear model for simple type cells. The convolutional method predicts simple and complex cell responses to natural image stimuli as well as grating tuning curves. The fitted models yield a continuum of values for the PReLU parameter across the sampled neurons, showing that the simple/complex nature of cells can vary in a continuous manner. We demonstrate that complex-like cells respond less reliably than simple-like cells. However, compensation for this unreliability with noise ceiling analysis reveals predictive performance for complex cells proportionately closer to that for simple cells. Most spatial receptive field structures are well fit by Gabor functions, whose parameters confirm well-known properties of cat A17/18 receptive fields.

**Funding:** This work was supported by the
Canadian Institutes for Health Research (https://
cihr-irsc.gc.ca/e/19.html, research grant MOP-
119498 to CB); and by the Fonds de Recherche du
Québec - Santé (https://frq.gouv.qc.ca/en/graduate
fellowship to JS). The funders had no role in study
design, data collection and analysis, decision to
publish, or preparation of the manuscript.

**Competing interests:** The authors have declared
that no competing interests exist.

## Author summary

Methods for recording increasingly many visual cortex neurons are advancing rapidly,
demanding new approaches to characterize diverse receptive fields. We present a compact
convolutional neural network model of early cortical neurons, which is uniformly applica-
ble to simple and complex cells, and whose parameters are straightforwardly interpretable
and readily estimated from responses to natural image stimuli. This novel approach intro-
duces a single estimated model parameter to capture the simple/complex nature of a neu-
ron's receptive field, revealing a continuum of simple vs complex-like behaviour. We
show that almost all complex-like cells exhibit a lower response reliability to repeated pre-
sentations of the same stimuli, compared to more reliable responses from simple-like
cells. Accounting for this "noise ceiling" brings predictive performance for complex cells
proportionally closer to that for simple cells. Using this model estimation approach with
natural images, we evaluate findings from previous approaches that were restricted to sim-
ple-type cells and the use of grating or white noise stimuli, revealing a diversity of Gabor-
like spatial receptive field shapes, which lie along a continuum of spatial bandwidths.

## Introduction

Neurons in the striate cortex are the first stage of visual processing to provide a rich represen-
tation of the elementary features needed to efficiently encode natural images [1]. Since the
work of Hubel and Wiesel [2] these neurons have been extensively studied with simple stimuli
such as bars, sinewave gratings, or white noise. However, predicting responses to arbitrarily
complex stimuli such as natural images requires quantitative models that are selective to both
location in visual space and stimulus features within those locations, with the model parame-
ters estimated using system identification [3–5]. Simple cells, a subset of striate cortex neurons,
have distinct excitatory and inhibitory receptive field regions with linear spatial summation,
and are phase-sensitive, i.e., give modulated responses to drifting gratings. Thus they are well
modeled by a linear-nonlinear (LN) model [6], which is relatively easy to experimentally esti-
mate. Complex cells, however, are phase-insensitive, and exhibit mixed excitatory/inhibitory
responses throughout the receptive field—they are generally thought to be integrating
responses from several nearby linear-nonlinear subunits, which are more difficult to estimate.
The task is further complicated by significant variability in responses of cortical neurons [7–9].

   The development of improved methods to quantitatively characterize the diverse receptive
fields of visual cortex neurons is becoming more important, in order to effectively exploit
newer data acquisition technologies that simultaneously record large numbers of single neu-
rons, such as two-photon imaging and high-density electrophysiology [10]. In such situations
the neurons may differ in multiple ways, which could render conventional tuning curve
approaches problematic.

   Receptive fields of simple type cells have been traditionally estimated by spike-triggered
averaging (STA) of responses to white noise (e.g. [11,12]). The nonlinear subunit structure of
complex cell receptive fields, however, has generally required more elaborate methods such as
spike-triggered covariance [12]. However, these methods produce more complicated models
(multiple eigenfunctions), requiring many more parameters than the linear-nonlinear model,
and are therefore harder to estimate and may give poorer predictive performance. The use of
different methods to analyze simple and complex cells is at odds with evidence indicating that
simple and complex cells may not be categorically distinct [13]. A combination of STA and
STC approaches can be used on both simple and complex cells [12,14], but the resultant

models are even more heavily parameterized, as well as being difficult to interpret. It would be highly desirable to have a method that works uniformly on simple or complex cells, or intermediate types, providing estimated models without over-parameterization.

Visual neurons also vary not only in their spatial properties, but also in their temporal dynamics—for example, sustained vs transient or lagged vs non-lagged behaviour, spatiotemporal non-separability, or temporal frequency response [15]. Many previous studies of visual cortex neurons employing system identification have only considered spatially 2D models, without time dynamics, or have included temporal properties at the expense of employing spatially 1-D stimuli (e.g., random bar stimuli) to estimate a "space-time" receptive field [16]. A full 3D analysis (space-space-time) has been challenging due to the much larger number of model parameters to be estimated. However, this would be most desirable for revealing the full spatiotemporal properties of receptive fields.

Most previous system identification methods for early visual neurons have necessitated the use of spatiotemporally uncorrelated ("white") stimuli. However, it is desirable to have methods that work for more complex stimuli such as natural images, which potentially provide results more relevant to normal operating ranges of image statistics [5,17]. Furthermore, neurons beyond the very earliest stages of visual processing may give poor responses to white noise and require complex stimuli to give responses more suitable for system identification [18,19]. It is possible to modify STA and STC-style approaches for responses to natural images [4,5,20,21]. Another approach involves estimating a general linear model that sums responses of a wavelet basis designed to mimic early-stage neurons [22,23], though this requires strong prior assumptions about the nature of the earlier-stage filters.

Recently, convolutional neural networks (CNNs) have become widely used in engineering/computer science applications due to their state of the art performance, particularly on image-related problems [24]. These models typically use a succession of layers, each having multiple spatially homogeneous convolutional filters, separated by half-wave rectifiers. Such multi-layer "deep" neural networks (DNNs) have been used to estimate responses of neurons in striate cortex [19,25–28]. However, it can be unclear how to interpret intermediate stages of such complicated model architectures. In addition, most such DNNs have previously been estimated only for 2-D spatial receptive field models (i.e., without time dynamics). To extend such approaches to full 3-D space-space-time models, it would seem problematic to estimate the multitude of parameters in such complex models using limited sets of data from the noisy responses of cortical neurons. Prenger et al [29] estimated 3-D spatio-temporal receptive fields using a feedforward neural network driven by inputs of pre-processed natural image stimuli (i.e., 25 main principal components of the stimuli). However, it would seem desirable to understand neural processing of raw natural image stimuli rather than their principal components. Furthermore, their model was challenging to optimize due to the need for elaborate regularization measures requiring many hyperparameters.

Here we present a compact convolutional neural network model, with parameters that are simple to interpret: a single initial 3-D spatiotemporal filter captures the feature selectivity, a single parameter explicitly models the simple/complex behaviour of the neuron (phase sensitivity), and a final layer localizes the filter activity (overall envelope of receptive field). The parameters of the model can be solved directly by gradient descent, using responses to natural images. The convolutional model prediction of neuron response performs well for most single neurons from cat A17 and A18, both for natural images and for tuning curves from sinewave gratings. The phase sensitivity estimates of the convolutional model trained on natural images is highly correlated with neurons' simple vs. complex type phase responses to sinewave gratings. Our analysis also shows that complex cell responses are inherently less reliable than those

for simple cells. Accounting for this "noise ceiling" brings predictive performance for complex cells proportionately closer to that for simple cells.

We also demonstrate how the approach can be used to measure cortical receptive field properties such as size and optimal spatial frequency, and how the distributions of these properties are related to simple vs. complex type responses and to the cortical area from which they are recorded. The results reveal receptive field shapes that may be Gabor-like, with diverse aspect ratios, as well as numerous instances of non-oriented receptive fields (e.g., centre-surround organization). We show how the method can be used to re-examine the finding [30] that these shape variations are constrained, such that they lie along a "main sequence" of varying tuning bandwidth.

## Materials and methods

### Ethics statement

All animal procedures were approved by the Animal Care Committee of the Research Institute, McGill University Health Centre, and are in accordance with the guidelines of the Canadian Council on Animal Care.

### Animal preparation

The procedures for animal surgery, recording, and maintenance were conventional and have been described in detail [31], so here they will be only briefly summarized. Anesthesia in adult cats of either sex was induced with isoflurane/oxygen (3–5%), followed by intravenous (i.v.) cannulation. Surgical anesthesia was then provided with i.v. propofol—first a bolus i.v. injection (5 mg·kg$^{-1}$), followed by a continuous infusion (5.3 mg·kg$^{-1}$·h$^{-1}$), and supplemental doses of propofol as necessary. Intravenous doses of glycopyrrolate (30 µg) and dexamethasone (1.8 mg) were provided, and a tracheal cannula or intubation tube was positioned for a secure airway. Body temperature was maintained with a thermostatically controlled heating pad, and heart rate was monitored (Vet/Ox Plus 4700).

The animal was then secured in a stereotaxic apparatus, and a craniotomy and small durotomy were performed over the cortical region of interest (A17, P3/L1, or A18, A3/L4), which was then protected by 2% agarose and petroleum jelly. All surgical sites were infused with local injections of bupivacaine (0.50%).

After completion of all surgical procedures, the animal was connected to a ventilator (Ugo Basile 6025) supplying oxygen/nitrous oxide (70:30). Additional anaesthesia was provided with i.v. remifentanil (bolus injection, 1.25 µg·kg$^{-1}$, followed by infusion, 3.7 µg·kg$^{-1}$·h$^{-1}$), with a reduced propofol infusion rate (5.3 mg·kg$^{-1}$·h$^{-1}$). Paralysis was then induced and maintained with i.v. gallamine triethiodide (bolus: to effect, followed by infusion: 10mg·kg$^{-1}$·h$^{-1}$).

Topical ophthalmic carboxymethylcellulose (1%) was used for protection of the corneas during initial surgery. Neutral contact lenses were later inserted for long-term corneal protection. Spectacle lenses were used to bring the eyes into focus at 57 cm, the distance at which visual stimuli would be presented. The optical quality was further improved with artificial pupils (2.5 mm). Topical phenylephrine hydrochloride (2.5%), to retract nictitating membranes, and a mydriatic (atropine sulfate, 1%, or cyclopentolate, 1.0%, in later experiments) were applied daily.

Additional doses of glycopyrrolate (16 µg) and dexamethasone (1.8 mg), intramuscular, were also provided daily. Recording experiments were conducted over a period of 3–4 days, during which vital signs (temperature, heart rate, expired $CO_2$ and EEG activity) were monitored and kept at normal levels.

## Visual stimuli and electrophysiology

Visual stimuli were presented on a gamma-corrected CRT monitor (NEC FP1350, 20 inches, 640x480 pixels, 150 Hz, 36 cd/m$^2$) at a viewing distance of 57 cm. Stimuli were produced by an Apple Macintosh computer (MacPro, 2.66 GHz, 6 GB, MacOSX ver. 10.6.8, NVIDIA GeForce GT 120) using custom software written in MATLAB (ver. 2012b) with the Psychophysics Toolbox (ver. 3.0.10; [32,33]). A photometer (United Detector Technology) was used to measure the monitor's gamma nonlinearity, which was corrected using inverse lookup tables. Signals from a photocell (TAOS, TSL12S) in one corner of the monitor were used for temporal registration of stimulus onset/offset timing and spike recordings, and to verify the absence of dropped frames.

Extracellular recordings were obtained via 32-channel multielectrode probes (NeuroNexus, linear array: A1x32-6mm-100-177, or polytrode: A1x32-Poly2-5mm-50s-177), and a Plexon Recorder data acquisition system. A single-channel window discriminator was used to isolate spiking responses from a single channel (300-5kHz) of interest for "on-line" monitoring during the experiment. However all 32 broadband (3Hz-5kHz) data channels (as well as CRT photocell signals, TAOS TSL12S) were streamed to hard disk for later off-line spike waveform classification [SpikeSorter: 34], and detailed analyses of each neuron's responses.

Electrodes were inserted in approximately vertical, near-columnar penetrations, so that neurons generally had similar receptive field location, preferred spatial frequency, and preferred orientation [2]. An interactively controlled bar stimulus was used to get an initial estimate of the neurons' receptive field location and dominant eye, from a representative channel. The non-dominant eye was occluded, and the CRT centered approximately on the receptive field.

For subsequent quantitative experiments, visual stimuli (gratings, natural images) were presented on the CRT display. The responses of one neuron on one channel were chosen to guide the optimization of grating stimuli. Drifting sinewave gratings were circularly cosine-tapered against a blank background at mean luminance. Tuning curve measurements were obtained from gratings presented in random order in 1 sec trials, with at least 10 repetitions of each condition. Only the background was presented for "blank" trials which were used to estimate spontaneous activity. Each neuron's "AC/DC ratio" (first harmonic divided by mean response) was calculated from the response to its optimal grating [35]. The grating orientation and temporal frequency were optimized specifically for each neuron, with a fixed value of contrast.

Natural image stimuli were constructed from monochrome 480x480 pixel subsets of photographs from the McGill Calibrated Color Image Database, https://pirsquared.org/research/mcgilldb/ [36]. Images having low pixel standard deviations were discarded, since they were nearly blank. The remaining images had their mean luminance subtracted and were RMS-normalized and clipped to 8 bits. These images were the same as used previously [18], but with a higher RMS contrast. Ensembles of 375 such images were presented in 5 sec trials, preceded by 100 ms of mean luminance blank screen in order to estimate spontaneous activity. Though the CRT refresh rate was 150Hz, each stimulus frame was presented twice for an effective rate of 75Hz. Stimulus image ensembles were divided into 3 sets, designed to be used for training, regularization, and testing (see below). There were 20 training ensembles (i.e., 7500 unique natural images), each presented 5 times. Both regularization and test sets each consisted of 5 ensembles (i.e., 1875 unique natural images for each) and were presented 20 times each. The same natural image ensembles were presented for all the neurons in the population, though some differed in contrast (as described above). The number of ensembles and repetitions were chosen as a trade-off between maximizing the distinct stimuli presented to the neuron while reducing the effect of trial-by-trial variability. Image ensemble sets were presented in pseudo-random order over a period of ca. 45 minutes.

## Model estimation

**Convolutional neural network (CNN) model.** To handle both complex and simple type cells, we introduce a "convolutional" model (Fig 1):

$$\hat{R}_t = N(\sum_{x,y} w[x, y] G(\sum_{j,k,\tau} c[j, k, \tau] s[x - j, y - k, t - \tau]))$$ (1)

In this model, the stimulus **s** is first spatiotemporally convolved with the filter **c**. The output of this convolution, at a given time $t$, is a two-dimensional image, which is then passed through a static nonlinearity $G$. It can be seen that at a fixed point (x, y), the convolution ($c^T s_t(x,y)$) is similar to the linear component of the linear-nonlinear model for a small region of the full stimulus. Similarly, $G(c^T s_t(x,y))$ can be thought of as similar to the linear-nonlinear model operating on a small region of the full stimulus. Note that the full convolution operation, $G(c^* s_t)$, applies the linear-nonlinear model to every possible $nxn$ sub-window of the stimulus, where $n$ is the size of the filter **c**. Thus, the operation $G(c^* s_t)$ can be considered as the activity of a homogeneous array of linear-nonlinear subunits in response to the stimulus $s_t$. Each sub-unit will have the same filter **c**, where the subunits are tiled across the stimulus space. The final set of weights, **w**, parameterizes the regions of visual space ("map", see below) where the sub-unit activity is predictive of the neuron's response **R**. The final output nonlinearity, $N$, is parameterized as a rectified power law exponent.

Instead of fixing $G(x)$ as a half-wave rectifier (as in traditional deep neural networks), we make a slight adjustment to allow for estimation of the linear-nonlinear model as well as the subunit model. We parameterize $G(x)$ as a "parameterized rectified linear unit" (PReLU), a nonlinearity that has been shown to improve the performance of artificial neural networks trained to classify images [37]. The PReLU is a simple static piecewise-linear rectifier:

$$G(x) = \begin{cases} x, & \text{if } x > 0 \\ \alpha x & \text{if } x \leq 0 \end{cases}$$ (2)

The PReLU has a single parameter $\alpha$ which denotes the slope of the negative component of the rectifier. If $\alpha = 0$, then $G$ acts as a half-wave rectifier; this would correspond to a

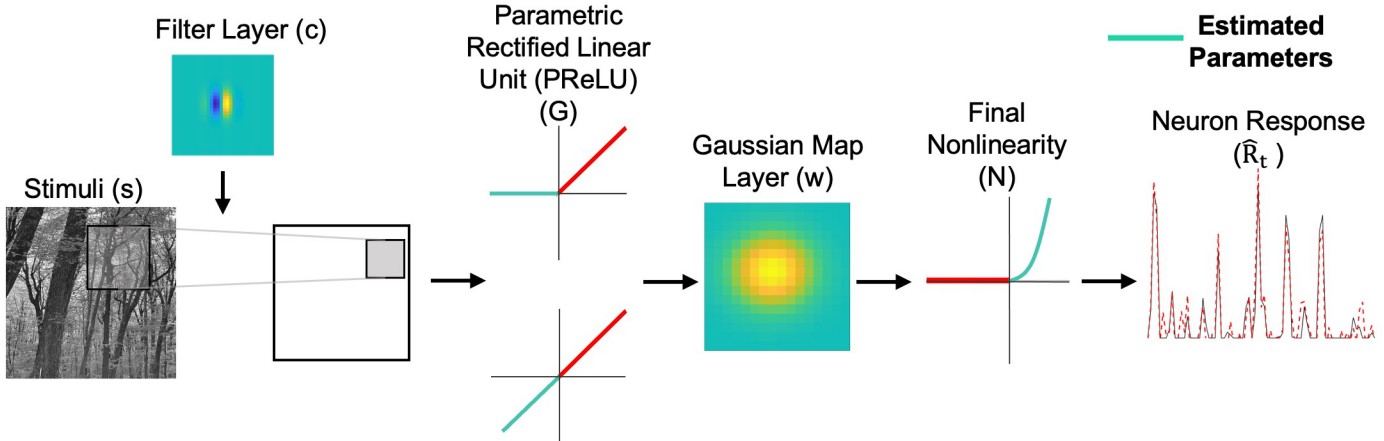

**Fig 1. Convolutional model for simple and complex type visual cortex neurons.** Stimulus frames are spatiotemporally convolved with the filter layer (*c*), then passed through a Parametric Rectified Linear Unit (PReLU) non-linearity (*G*). The dot product between the PReLU layer output and the spatial map (*w*) layer generates the neuron response. A final rectified power law nonlinearity (*N*) is added to account for the spiking threshold and possible nonlinear spike frequency response. Cyan elements represent estimable parameters. To avoid copyright issues, the depicted natural image ("Stimuli") was not one of those used experimentally.

conventional complex cell instance of the subunit model as mentioned above. If $\alpha = 1$, then $G$ is the identity function ($G(x) = x$) and the linear weights of both $w$ and $c$ will be in a linear cascade, which is equivalent to a single linear layer—thus the convolutional model then collapses to a linear-nonlinear model conventionally used for a simple cell.

The nonlinearity $G$ could be modeled more generally, for example as a set of piecewise linear B-spline ("tent") basis functions [38], which provide a local approximation of high order polynomials. However, when using a tent basis, G cannot be estimated concurrently (i.e., within the same gradient descent) as the convolutional filter $c$ and map weights $w$. This problem can be solved by using an EM-like algorithm such as alternating least squares [38,39], that alternately keeps one part fixed while optimizing the other. However, such methods take longer to calculate, might incur a greater risk of convergence on local optima, and may be less robust with choice of initial conditions. Modelling $G$ as a PReLU allows us to estimate $\alpha$ jointly with the other parameters of the model, since it can be incorporated into the same gradient descent backpropagation that estimates the filter and map functions.

We refer to $w$ as the "map" layer, because it indicates the region in visual space where the subunit filter $c$ is operative. Prior knowledge of the receptive field shape can be used to define model parameters [40]. The map layer embodies the spatial envelope of the neuron's receptive field, which for early visual cortex neurons can be parameterized as a two-dimensional Gaussian. Thus the map layer $w$ can be defined with only six parameters—two for the means ($\mu_X$, $\mu_Y$), three for the variances (covariance matrix $\Sigma$), and a scale factor $\beta$:

$$w[x,y] = \beta \cdot \eta_{\mu,\Sigma}(x,y) = \beta \cdot \frac{\exp(-1/2(x - \mu_X; y - \mu_Y)\Sigma^{-1}(x - \mu_X; y - \mu_Y))^T}{2\pi\sqrt{|\Sigma|}} \tag{3}$$

We also estimate the $N$ power law exponent in the final layer of the model. $N$ is set to be an adaptive exponent output nonlinearity defined in a custom Keras layer (see below) with two trainable parameters gain (g) and power law exponent (exp) as:

$$N(x) = \begin{cases} g\,x^{exp} & if\ x > 0 \\ 0 & if\ x \leq 0 \end{cases} \tag{4}$$

## Estimation of model parameters

Data files were converted from the Plexon file format (.plx) to MATLAB data files (.mat) using in-house code. Spikes were sorted [34] independently for each stimulus set (i.e. natural image ensembles, tuning curves for different grating parameters). Sorted waveforms were then matched (Euclidean similarity) to establish corresponding neurons across the two kinds of stimulus sets. This approach was chosen over concatenating responses from gratings and natural image movies prior to spike sorting, to minimize effects of possible probe drift between natural image and grating sessions, particularly since these responses were often collected at non-contiguous times. Only matches with a high degree of confidence were accepted for use in those analyses requiring comparisons of responses to different stimuli. The neuron response was averaged over the repetitions of each natural movie stimulus ensemble or grating parameter value.

In methods for estimating receptive field models from responses to complex stimuli which could be much larger than the receptive field, it is important to first "crop" the stimulus images to include only the parts that encompass the receptive field [17], prior to further analysis. This procedure is necessary to avoid using too many pixels, that can result in overparameterization,

i.e. estimating unnecessary parameters, leading to a loss in accuracy of the estimated model. Here we employ a three-pass cropping procedure. In the first pass, we estimate the model parameters using the full 480x480 stimulus images downsampled to 30x30. We then manually identify a square-shaped cropping window that encloses an area slightly larger than the apparent receptive field, and downsample the image within it to 30x30. This cropped image is then used to re-train the model in the second pass. In the third pass, we adjust the cropping window based on the model estimate obtained in the second pass to identify accurate boundaries of the receptive field. This cropping procedure will be illustrated below in Results for an example neuron. To handle very small receptive fields, we imposed a minimum cropping window size of 120 pixels. However, the majority of A18 neurons and a subset of A17 neurons having large receptive fields required only the first one or two passes.

The amount of downsampling employed after cropping was then determined by the choice of filter size—if the filter dimension is too low, the estimated filters are too small, noisy, and may appear clipped off at the edges. On the other hand, an overly large filter size can result in overparameterization with resultant loss of predictive performance. Preliminary analyzes indicated best results with a filter size of 10 to 15. We evaluated for each neuron, filter sizes of 11 and 15 pixels, and chose the one giving filters which exhibit minimal clipping of the edges while providing a higher predictive accuracy (VAF) for the validation dataset.

Simultaneously estimating the final output nonlinearity $N$ with other model parameters using backpropagation is problematic due to "gradient explosion". Therefore, $N$ is estimated in two stages. In the first stage, $N$ is set to be a simple ReLU (half-wave rectifier) activation function and an initial estimate of the receptive field and other model parameters are obtained. In the second stage, the pre-trained receptive field weights are loaded, and the model is re-trained with $N$ set to be an adaptive exponent output nonlinearity as defined in Eq 4. The total number of model parameters estimated for each neuron was 855 for filter size 11 (filter: 11 x 11 x 7 + PReLU: 1 + gaussian map: 5 + output nonlinearity: 2) and 1583 for filter size 15 (filter: 15 x 15 x 7 + PReLU: 1 + gaussian map: 5 + output nonlinearity: 2).

We solve for all parameters in the model with gradient descent, using a loss function of the mean square error between the neuron's measured response and the model's predicted response for the training dataset. Bias terms ("dc offsets") are included where appropriate. The gradient of the error is backpropagated in order to estimate each parameter in each layer. Optimization of the convolutional model was implemented in Python using tools from the Tensor-Flow optimization package, version 2.0.0 [41] and Keras, version 1.1.2 [42]—see https://www.tensorflow.org/ and https://keras.io/. TensorFlow is used for its ability to automatically compute the gradient with respect to each parameter, including the $\alpha$ parameter of the PReLU and the Gaussian parameters of the map layer, as well as its built-in support for GPU processing. The optimization was performed using the Adam optimizer [43], and with early stopping regularization using a "patience" parameter of 50. Additionally, L2 penalty regularization was applied to the filter weights—we evaluated a range of L2 hyperparameter values, however the model performance was maximum at 0.01 which is the default for a Keras conv2D layer—larger L2 hyperparameter values also caused excessive smoothing in estimated filter weights and lower model performance accuracies. The SciPy/NumPy Python packages are also extensively used. The weights for the filter are initialized randomly [44], with a taper applied at the edges to encourage non-zero weights towards the center of the filter array. The PReLU $\alpha$ is initialized at 0.5, and the Gaussian map layer is initialized to the center of the image, with standard deviation equal to the length of the image (after cropping and downsampling, i.e., 30 pixels).

Code for a working example of the model estimation method can be found at our GitHub (https://github.com/JinaniSooriyaarachchi/CNNwithPReLU_RFestimateV1).

## Model validation

The parameters of the model are optimized using the training dataset, with early stopping based on predictive ability for the regularization set. The test set is held out until all training procedures are completed, to give an unbiased estimate of the models' predictive accuracy. The model accuracy is measured by the variance accounted for (VAF), which is computed as the square of the correlation between the neuron's measured response and the response predicted by the estimated model. This "raw" VAF (averaged over all 20 repetitions of the test set, and expressed as a percentage), however, does not take account of degradation by noisy responses of the neuron.

To measure the trial-by-trial variability of a neuron's response, we estimate a "noise ceiling" as follows. For each of the 20 repetitions of the test set, the neuron's response on that repetition is used to predict the average response for the 19 other repetitions of the same stimulus, and a VAF ($R^2$) is calculated for this prediction. The average of these 20 VAFs provides an estimate of the noise ceiling, i.e., an upper bound on the explainable variance in the neuron's response, $R^2_{neuron}$. This approach uses the average of the 19 other repetitions to reduce the noise greatly (if not entirely) for use as a "ground truth". Our approach of calculating the noise ceiling value is similar to the "oracle score" used by previous studies [38,45].

The model's performance is computed similarly to $R^2_{neuron}$. For each of the 20 repetitions of the test set, the neuron's response on that repetition is compared with the model's prediction of the neuron response, and a VAF ($R^2$) is calculated for this prediction. The average of all 20 of these VAF values is denoted as $R^2_{model}$. The ratio between $R^2_{model}$ and $R^2_{neuron}$ is an estimate of the proportion of "explainable" variance achieved by the model [46]. This value, again expressed as a percentage, is used to quantify a "noise-corrected" estimate of the model performance.

## Simple-complex measure

For purposes of comparison with the estimated PReLU α values, the ratio of first harmonic to mean response to an optimized grating, or "AC/DC" value, has been widely used to classify a cortical neuron as a simple or complex type cell [35]. Neurons with an AC/DC less than unity were classified as complex, otherwise they were classified as simple. We calculated the AC/DC value for neurons on which sinewave grating responses were collected, and correspondence of sorted spike waveforms could be established (see above).

## Analysis of spatial receptive field properties

In the convolutional analysis, both the filter and map layers contribute to the spatial receptive field properties of a neuron. For example, when the filter has oriented, alternating excitatory and inhibitory regions and the map function is an isotropic Gaussian, the optimal orientation will be determined by the filter, but the overall size of the receptive field will depend jointly on both the filter and map functions. For archetypal simple type cells (alpha = 1.0) which can be modeled as a linear filter followed by a pointwise nonlinearity ("LN" model), the spatiotemporal receptive field can be "reconstructed" by taking the convolution of the filter function **c** with the map function **w**. The phase-invariance of complex cells precludes a simple LN model, however we note that the PReLU can be considered as a weighted sum of a linear function and a full-wave rectifier, which give linear and nonlinear responses, respectively, analogously to first and second Volterra kernels [47]. For the linear part, the PReLU contributes only a scale factor, and again we can take the convolution of the filter function with the map function ("restoration") to provide a first-order indication of the receptive field properties, analogous to a first Volterra kernel. Here we will examine the restoration functions, which provide a consistent

and uniform approach to characterizing properties of both simple and complex type cells, as well as those having intermediate properties, without having to treat them as categorically distinct.

We analyze spatial receptive field properties by considering the restoration at the time lag having maximal variance [30,48]. This approach was used because many A17 and A18 neurons exhibit a substantial amount of space-time non-separability [11,18,49–51]. To characterize some of the conventional parameters of visual receptive fields, we find a best-fitting Gabor function (Eq 5, below) for each RF restoration. These functions fit most receptive fields well [52,53], provide parameters of interest directly in the fitted function, and also have a rich theoretical underpinning [53]. We parameterize the fitted Gabor function as a sinusoid multiplied by a Gaussian envelope, with a change of variable to handle orientation:

$$h(x', y') = A \exp\left(-\left(\frac{x'}{\sqrt{2}\sigma_{x'}}\right)^2 - \left(\frac{y'}{\sqrt{2}\sigma_{y'}}\right)^2\right) cos(2\pi f x' + \varphi) + d \qquad (5)$$

where

$$x' = (x - x_0)cos\theta + (y - y_0)sin\theta$$

$$y' = -(x - x_0)sin\theta + (y - y_0)cos\theta$$

There are 9 free parameters, whose values are fit to a given spatial filter:

A is the overall amplitude or gain

f is the spatial frequency of the sinusoid

$\theta$ is the orientation of the sinusoid

$\varphi$ is the phase of the sinusoid

$\sigma_{x'}$ and $\sigma_{y'}$ are the standard deviations of the Gaussian along orthogonal axes, i.e., width and length respectively

$x_0$ and $y_0$ are the center location

d is an offset

To find values of the parameters that minimize the summed square error between a measured RF and fitted Gabor, we employ a modified version of Kendrick Kay's *fitGabor2d* and its associated functions (*https://github.com/kendrickkay/knkutils*). This code uses Matlab's *lsqcurvefit* function which finds best-fitted parameters in the least-square sense. We modified this code to avoid falling into local minima during fitting, by evaluating results from multiple different initial values for the phase parameter, $\varphi$, to obtain the best fit.

Goodness of fit is indicated by the fraction of variance unexplained, FVU [30]:

$$FVU = 1 - R^2 \qquad (6)$$

where $R^2$ is the coefficient of determination. FVU was chosen as a measure here, to avoid possible confusion with the VAF measure of predictive performance, and for consistency with Ringach et al [30]. FVU ranges from 0 to 1.0, with decreasing values indicating better fits. As will be seen below, in the majority of cases the fitting algorithm converged and gave good fitting functions that corresponded well with the spatial receptive fields.

## Analysis of temporal receptive field properties

Our approach estimated spatial receptive fields of neurons across seven time lags to also capture their temporal dynamics. Using the spatial receptive field (restoration) at the time lag having maximal variance, we identified a 3 x 3 pixel region around the (x,y) location with the

highest magnitude. We then extracted the average of these 9 pixels for each of the measured time lags, to provide a temporal response function. Since these temporal functions typically resembled a damped sinusoid, we fit a damped sine wave function to these seven time points (Eq 7):

$$y(t) = Ae^{-\lambda t}\cos(\omega t + \phi) \tag{7}$$

where

A is the amplitude
$\lambda$ is the (reciprocal) decay constant
$\omega$ is the angular frequency
$\phi$ is the phase angle.

We assessed the quality of the damped sine wave fit by calculating the FVU (as above, Eq 6). Using $\omega$ from the fitted damped sine wave, we obtained an estimate of the optimal temporal frequency ($f = \frac{\omega}{2\pi}$) for each neuron.

## Results

A total of 271 neurons were isolated from multielectrode recordings in 12 cats (some of which also contributed to recordings that were part of other concurrent studies). Of these, 206 neurons (127 from A17 and 79 from A18) exhibited sufficient response reliability ($R^2_{neuron} > 1.0\%$ for both the regularization and test datasets) to be included. Neurons with an explainable VAF less than 1.0% were also excluded, to help ensure reliability of the estimated model parameters. These rather low exclusion thresholds were chosen because we noted that we typically obtained well-structured RF estimates even for these lower VAF and $R^2_{neuron}$ values. In addition, we also required that the neurons' average firing rates in response to visual stimuli were at least 0.5 spikes/second greater than to the blank controls. The accepted 168 neurons (104 from A17 and 64 from A18) were used in further analysis as explained in the following sections.

### Convolutional model predictive performance

The three-pass cropping procedure played an important role in characterizing a neuron's receptive field with high resolution and substantially increased predictive performance. Detailed results from an example neuron are shown in Fig 2. In the first pass, the estimated receptive field filter has poor resolution, the gaussian map layer is confined within a single pixel, and the linear restoration reveals little detail—such results are unsatisfactory for characterizing receptive field properties, because the lack of cropping leads to inappropriate downsampling. The estimated filter and map layer in the second pass are somewhat better with an acceptable linear restoration for the receptive field. However, estimated weights of a third pass yield a much better resolution of the structure of both the filter and map layers, and a more satisfactory linear restoration (Fig 2A and 2B). The VAF accuracy on a hold-back test dataset for this neuron improved from 46.2% in the first pass, 58.1% in the second pass to 61.3% in the third pass. An example stimulus image along with the cropping windows for this neuron (red and green boxes) are shown in Fig 2C. Fig 2D illustrates the overall improvement of VAF accuracy across three passes for a subset of A17 neurons which had sufficiently small receptive fields to be mapped through all the three passes (n = 83; mean VAFs: 15.1% in pass 1, 26.8% in pass 2 and 33.9% in pass 3). Since the VAF improvements for most neurons after pass 3 were diminishing, we decided to choose pass 3 as the final pass for most of the neurons—however for neurons with larger RFs, only one or two passes might be sufficient. The final-pass cropping was qualitatively assessed to ascertain that it captured the RF fully (S1 Fig). Consequently, the remainder of the paper will present results from the final pass.

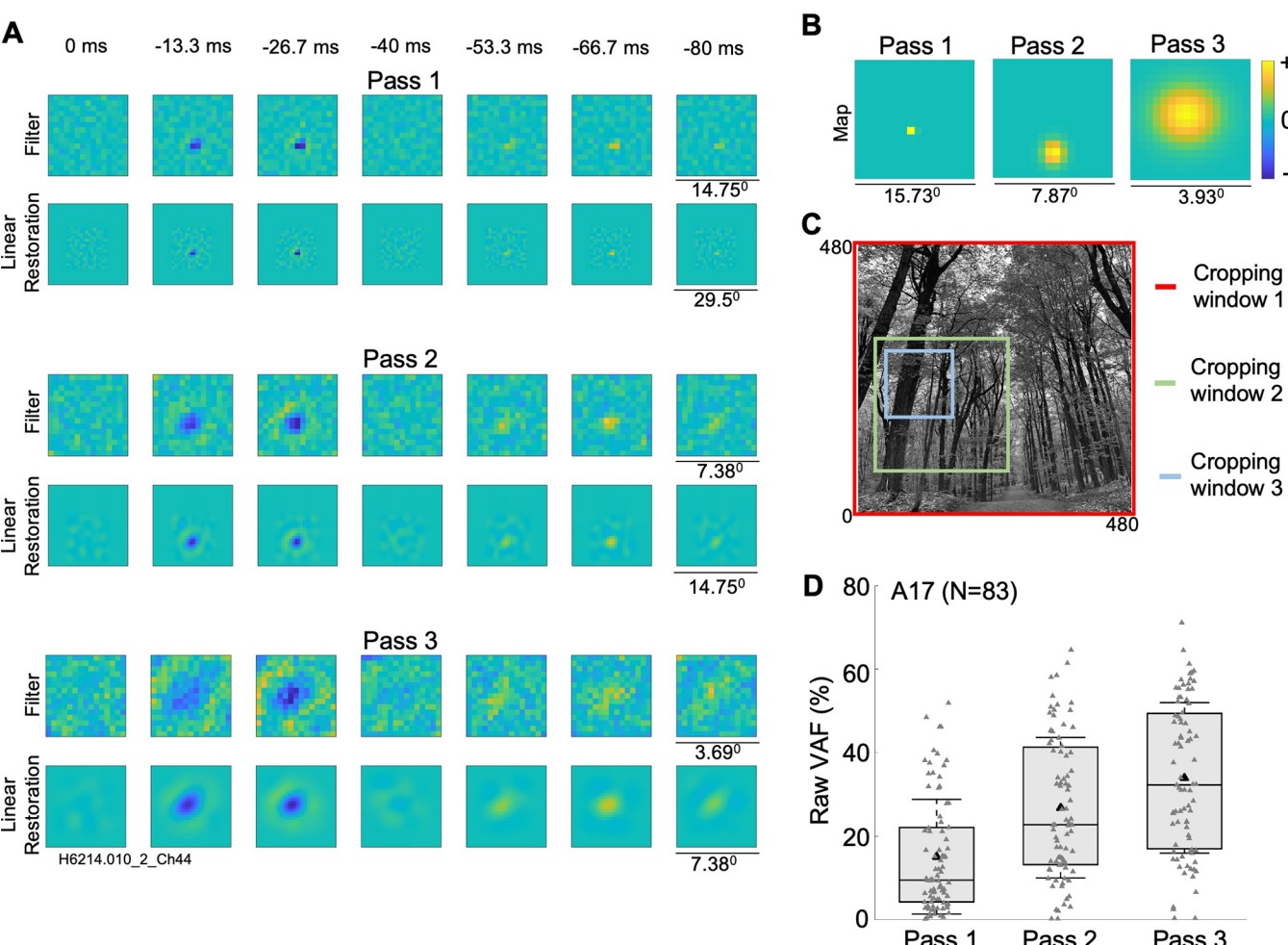

**Fig 2. Three pass procedure to estimate a better cropping window for an example cell from A17.** (A) Estimated receptive field filter and linear restoration (convolution of filter with gaussian map) for three passes with cropping windows of 480x480 in first pass (VAF = 46.2%), 240x240 in second pass (VAF = 58.1%) and 120x120 in third pass (VAF = 61.3%), each downsampled to 30x30. Note that the linear restoration dimensions (30 x 30) are larger than the filter dimensions, due to the convolution of the filter weights (15 x 15) with the gaussian map (16 x 16), here and in subsequent Figs 3 and 4. (B) Map layer for three passes, estimated as a parameterized 2D Gaussian. (C) Example cropping windows from one neuron (pass1: 480 x 480; pass2: 240 x 240; pass3: 120 x 120) overlaid on an example natural image stimulus frame. To avoid copyright issues, the depicted natural image was not one of those used experimentally. (D) Model VAF improvement for a subset of A17 neurons with sufficiently small receptive fields (n = 83) across three passes. Distributions indicated as box and whisker plots, in which black triangles indicate mean values, whiskers indicate standard deviations, box bounds indicate 25th and 75th percentiles, and horizontal lines indicate medians.

An example of detailed results from a simple-type cell are shown in Fig 3. The estimated convolutional model performed well, with a VAF of 61.3%. Its estimated receptive field filter (Fig 3A) contained a central inhibitory region and surrounding excitatory region. The map layer (Fig 3C), parameterized as a Gaussian, localizes the activity of the filter. In this example, the input image of 30x30 is convolved with a filter of 15x15 to produce a map layer of 16x16 pixels. In general the size of the filter layer was treated as a hyperparameter whose values were optimized for each neuron—in general the size of the map layer depends on the size of the filter due to the convolution. The PReLU (Fig 3B) had an α of 0.8, corresponding to an approximately linear relationship. Since α is close to 1.0 ($G(x) = x$) for this neuron we can simply neglect the PReLU and reconstruct the receptive field by convolving the filter with the map layer. Fig 3E shows the resulting linear receptive field restoration for this neuron. This restoration appears less noisy and has a more clearly delineated excitatory surround. Note that the

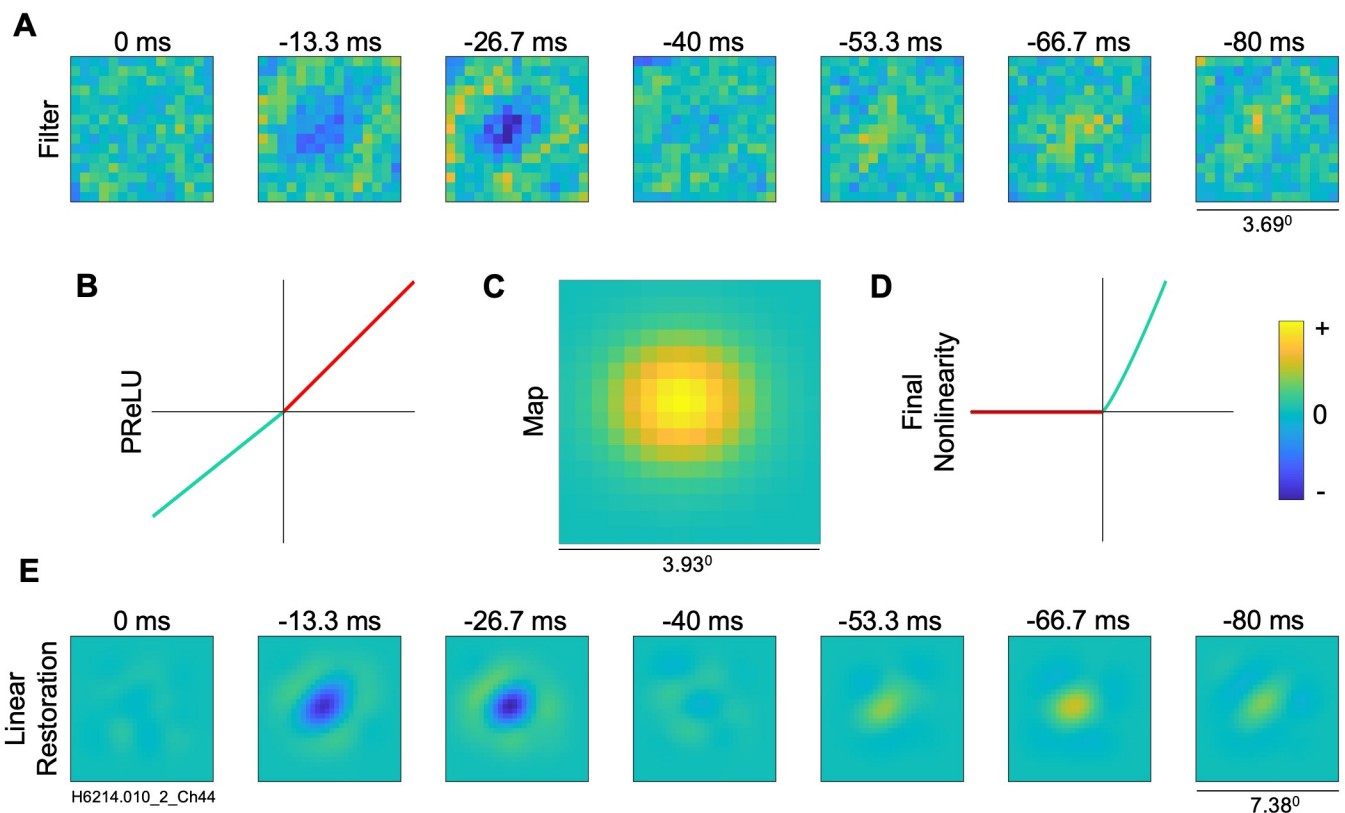

**Fig 3. Visualization of models estimated for an example simple cell from A17.** (A) Filter layer, 15x15 pixels. (B) Intermediate nonlinearity (parameterized rectifier unit, PReLU) between filter and map layers, having a fixed positive unity gain for positive inputs (red) and an estimated gain parameter, α, for negative inputs (cyan); here, the estimated α = 0.8. (C) Map layer, estimated as a parameterized 2D Gaussian; note different spatial scale. (D) Final output nonlinearity (exponent 1.2). (E) Linear filter restoration for the convolutional model (30x30 pixels), from convolution of filter and map layers. The convolutional method gave a raw VAF of 61.3%.

convolutional model architecture implicitly embodies an important constraint, i.e., that signal filtering properties of the neuron are determined by a relatively small filter, which is operative on only a limited subset (map layer, receptive field) of the input image. The Gaussian map layer applies a constraint on the locality of the filter activity, which reduces the noise outside of the neuron's receptive field. These factors together result in the linear restoration (Fig 3E) having smoother excitatory and inhibitory regions, on a background that is less noisy. The convolutional output from the map layer passes through the final output nonlinearity (exponent 1.2; Fig 3D) to generate the model's predicted neural response.

Results of the convolutional model parameters estimated for an example complex type cell are shown in Fig 4, giving a raw VAF of 15.1%. The filter layer (Fig 4A) is noisy compared to the above example simple cell filter, but nevertheless shows a vertically oriented inhibitory zone in the third time lag. The PReLU (Fig 4B) has an α = -0.12, indicating a substantial nonlinearity and thus a large deviation from the linear-nonlinear model conventionally used for simple-type cells. The notions often used for complex cells are more appropriate here: the filter function can be considered a subunit, with multiple such (rectified) subunits contributing to the response of the neuron, as proposed by Hubel and Wiesel [2]. Considering the PReLU as a weighted sum of a linear proportionality and a full wave rectifier corresponding to first and second order Volterra kernels (see Methods), the linear restoration of the complex cell (Fig 4E) provides a first

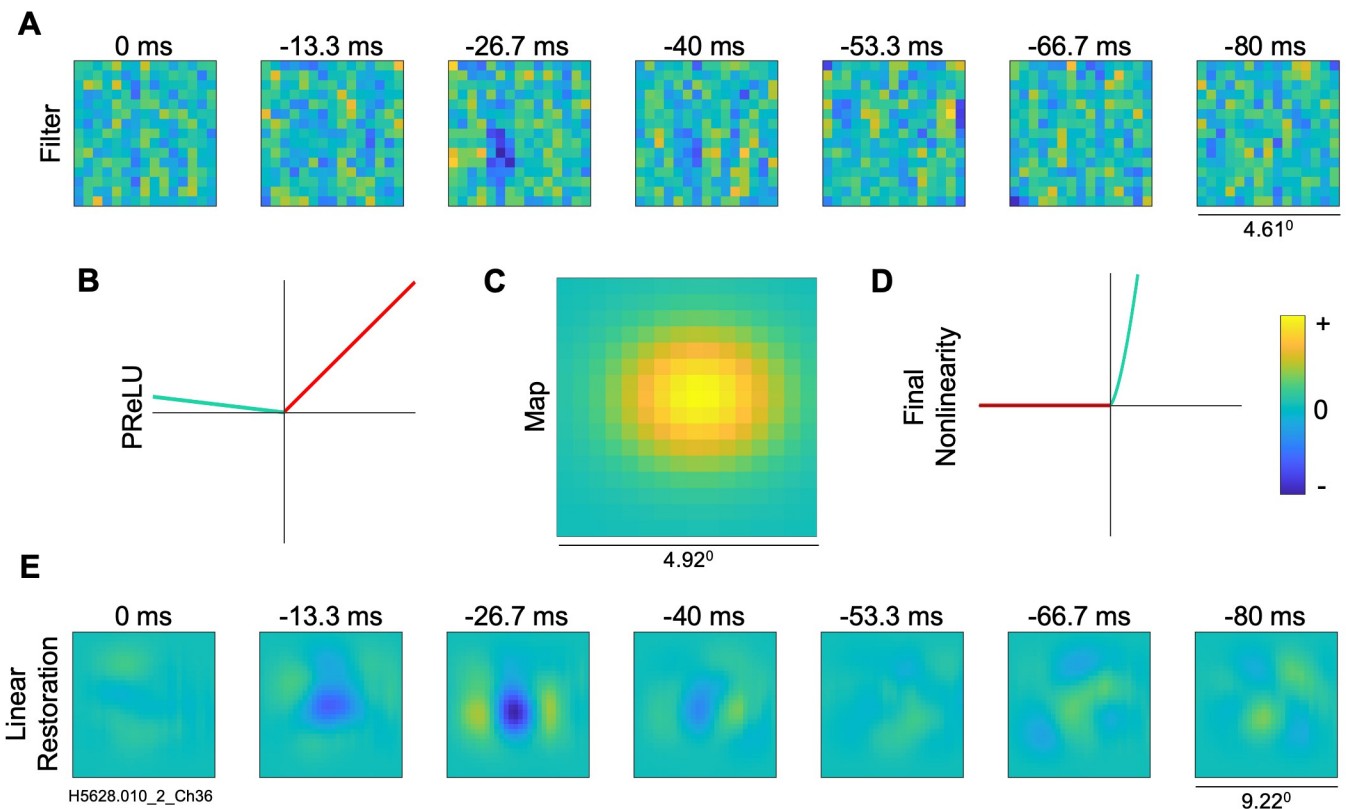

**Fig 4. Visualization of model estimated for an example complex cell from A17.** (A) Filter layer, 15x15 pixels. (B) PReLU nonlinearity, illustrated as for Fig 3, here with estimated α = -0.12. (C) Map layer (D) final output nonlinearity (exponent of 1.5) (E) Linear filter restoration (30x30 pixels), from convolution of filter and map layers. The convolutional model gave a raw VAF of 15.1%.

order indication of the receptive field properties. The final output nonlinearity (exponent of 1.5) for this complex cell example is shown in Fig 4D.

Across the sample of neurons, there was a very wide variation in predictive performance (Fig 5A), with VAF values ranging from near zero to ca 70% in both A17 and A18 neurons. The explainable VAFs, which adjusted for the "noise ceiling" in each neuron, showed significantly higher values compared to raw VAFs (Fig 5A, Mann-Whitney U test A17: p = 1.9e-08, n = 104, A18: p = 9.9e-05, n = 64). Previous system identification [14] generally found much lower VAFs for complex than for simple type cells. Since our sample includes a mixture of both kinds of cells, it then seems likely that many or most of the neurons with lower VAFs in Fig 5A might be complex cells.

This matter was addressed by comparing the raw and explainable VAFs for a subset of neurons whose AC/DC values were obtained from grating responses (Fig 5B). The raw VAFs for simple cells (AC/DC >= 1.0, n = 21) were almost always greater than those for complex cells (AC/DC < 1.0, n = 8). The simple cells had an average raw VAF of 44.3% whereas the complex cells had an average raw VAF of 21.9% (p = 0.0047, significant by a two-sample t-test). The explainable VAFs were significantly higher for complex cells compared to their raw VAFs (Fig 5B, Mann-Whitney U test A17: p = 0.03, n = 8). This effect was also observed for simple cells (Fig 5B, Mann-Whitney U test A17: p = 0.01, n = 21). The simple cells had an average exp VAF of 55.3% whereas the complex cells had an average exp VAF of 41.4% (p = 0.044, significant by Mann-Whitney U test). However, the difference between simple vs. complex mean exp

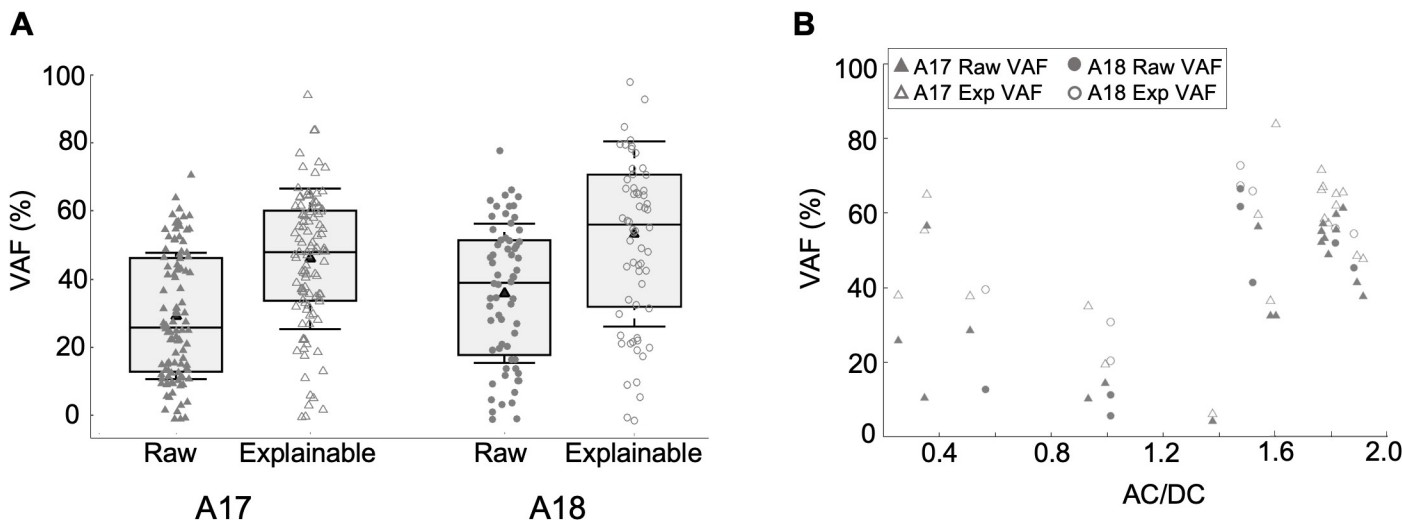

**Fig 5. Predictive accuracy for a holdback test dataset.** (A) Percentage of variance accounted for (raw VAF) in neurons' responses for A17 (mean VAF = 30.2%, n = 104) and A18 neurons (mean VAF = 36.5%, n = 64), respectively, and the explainable variance (exp VAF) in neurons' responses for A17 (mean VAF = 46.5%, n = 104) and A18 neurons (mean VAF = 53.9%, n = 64), respectively. Distributions indicated as box and whisker plots, in which black triangles indicate mean values, whiskers indicate standard deviations, box bounds indicate 25th and 75th percentiles, and horizontal lines indicate medians. (B) Comparison between the model prediction accuracy (raw VAF and exp VAF, closed and open symbols, respectively) and the neuron's phase sensitivity (AC/DC) to sinewave gratings (n = 29).

VAFs (13.9%) is relatively lower (22.4%) compared to the difference between raw VAFs. Therefore, adjusting for noise ceiling removes much of the difference in predictive accuracies for simple vs. complex-like cells.

### Prediction of grating responses

We additionally assessed the convolutional model by measuring how well it predicts neurons' responses to grating stimuli, for those neurons on which both kinds of responses were available. These models were estimated from the responses to the natural image stimuli, then simulations of the models were used to predict their responses to grating stimuli. The quality of prediction is assessed by the VAF between the predicted spatial frequency tuning curve and the neuron's measured tuning curve.

An example of such a comparison for spatial frequency tuning is shown in Fig 6A for an example A17 simple cell (more example tuning curves: S2 Fig). Here the neuron's measured response (black) and the convolutional model prediction (red) are quite similar (VAF = 91%), with clear bandpass tuning to about 1.0 cpd. Results for spatial frequency tuning prediction in 39 neurons (21 from A17, 18 from A18) are shown in Fig 6B. Considering the neurons in both brain areas together, the convolutional model performed with an average VAF of 58.2%. Considering only the A18 neurons, the average VAF from the convolutional model is 51.3%, and for the A17 neurons, the average VAF for the convolutional model is 64.1%. The model performance in predicting optimal spatial frequency was independent of the grating-derived AC/DC ratios (Fig 6C, p = 0.5, r = -0.2, n = 39), i.e. similar for simple- and complex-like cells. In summary, the convolutional model is reliable in predicting spatial frequency tuning curves for both A17 and A18 neurons.

### Noise ceiling analysis and simple vs. complex cells

Lower VAFs for complex cells could be expected since inadequacies of the associated model architecture might be greater than for simple cells. However, another possibility is that

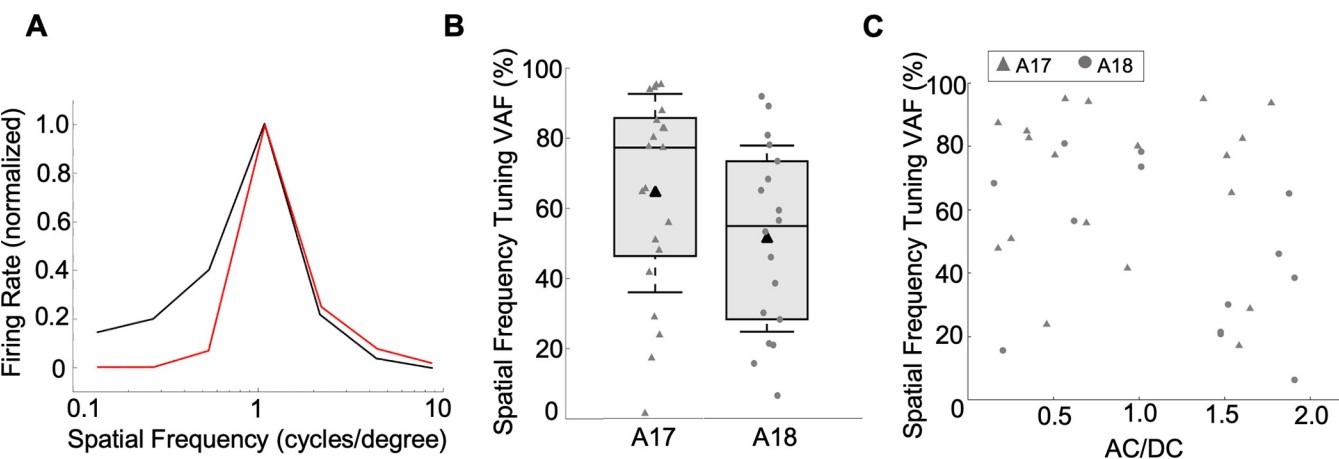

**Fig 6. Prediction of grating tuning curves by estimated models.** (A) Spatial frequency tuning curve (average firing rate, normalized to maximum) for an example A17 neuron. Temporal frequency, 2 Hz; average of 10 repetitions of 1 sec each. Solid black line for neuron's response to the grating stimuli, red for tuning curve predicted by convolutional model (VAF = 91%). (B) Model performance (VAFs) for spatial frequency tuning predictions (n = 18 for A18, n = 21 for A17). Distributions indicated as box and whisker plots, in which black triangles indicate mean values, whiskers indicate standard deviations, box bounds indicate 25th and 75th percentiles, and horizontal lines indicate medians. (C) Model performance (VAFs) for spatial frequency tuning predictions (n = 18 for A18, n = 21 for A17) and the neuron's phase sensitivity (AC/DC) to sinewave gratings.

responses of simple cells might be less noisy, i.e. exhibit less trial to trial variability, than for complex cells. We examined this idea using neurons in our sample for which we had responses to gratings as well as natural images. Fig 7A shows raster plots of neuronal firing across repeated trials in response to a subset of the natural image ensembles, for example neurons identified as simple or complex type cells based on their AC/DC ratios for optimized gratings. It can be seen that the raster points for the example simple cell (top) are relatively well aligned vertically, indicating fairly reliable responses, while the complex cell (bottom) gives much more variability in spike firing times across repetitions of the same stimulus ensemble. We quantified the trial-wise reliability by measuring how well the response on a given trial was correlated with the average response across all the repeated trials, using the $R^2_{neuron}$ index (see Methods)—higher values of $R^2_{neuron}$ indicate greater reliability. For the example neurons in Fig 7A, $R^2_{neuron}$ was 59.4% for the simple cell and 3.1% for the complex cell.

We similarly measured $R^2_{neuron}$ values from natural image responses for all of the sampled neurons for which responses to gratings were also available. Neurons with $R^2_{neuron}$ less than 1.0% on either the natural stimuli datasets (regularization and test), or less than 1.0% on the preferred grating stimuli (see below), were excluded. Results for 29 neurons (20 from A17, 9 from A18), shown in Fig 7B, reveal generally greater reliability in the simple cells (AC/DC > = 1.0) than in the complex cells (AC/DC < 1.0). For neurons classified as simple, the average $R^2_{neuron}$ (34.59%, n = 21) was significantly greater (t-test, p = 0.0105) than for neurons classified as complex (average $R^2_{neuron}$ = 14.56%, n = 8)—this difference is illustrated in the bar and whisker plots on the left side of Fig 7D. The $R^2_{neuron}$ is higher for responses to natural images compared to gratings (Fig 7D). This difference could be due to the natural image movie's temporally abrupt changes which might facilitate synchronization of the responses [54].

Conceivably this difference in reliability between simple and complex cells might be peculiar to their responses to natural images. To evaluate this idea, we performed the same analysis on responses to drifting sinewave gratings across repeated trials. Since this analysis required only grating responses, we were able to utilize a much larger sample of 100 neurons (73 from A17, 27 from A18). Cells with $R^2_{neuron}$ less than 1.0%, were removed to ensure valid AC/DC measurements. Fig 7C shows these neurons' $R^2_{neuron}$ values for responses to their optimized

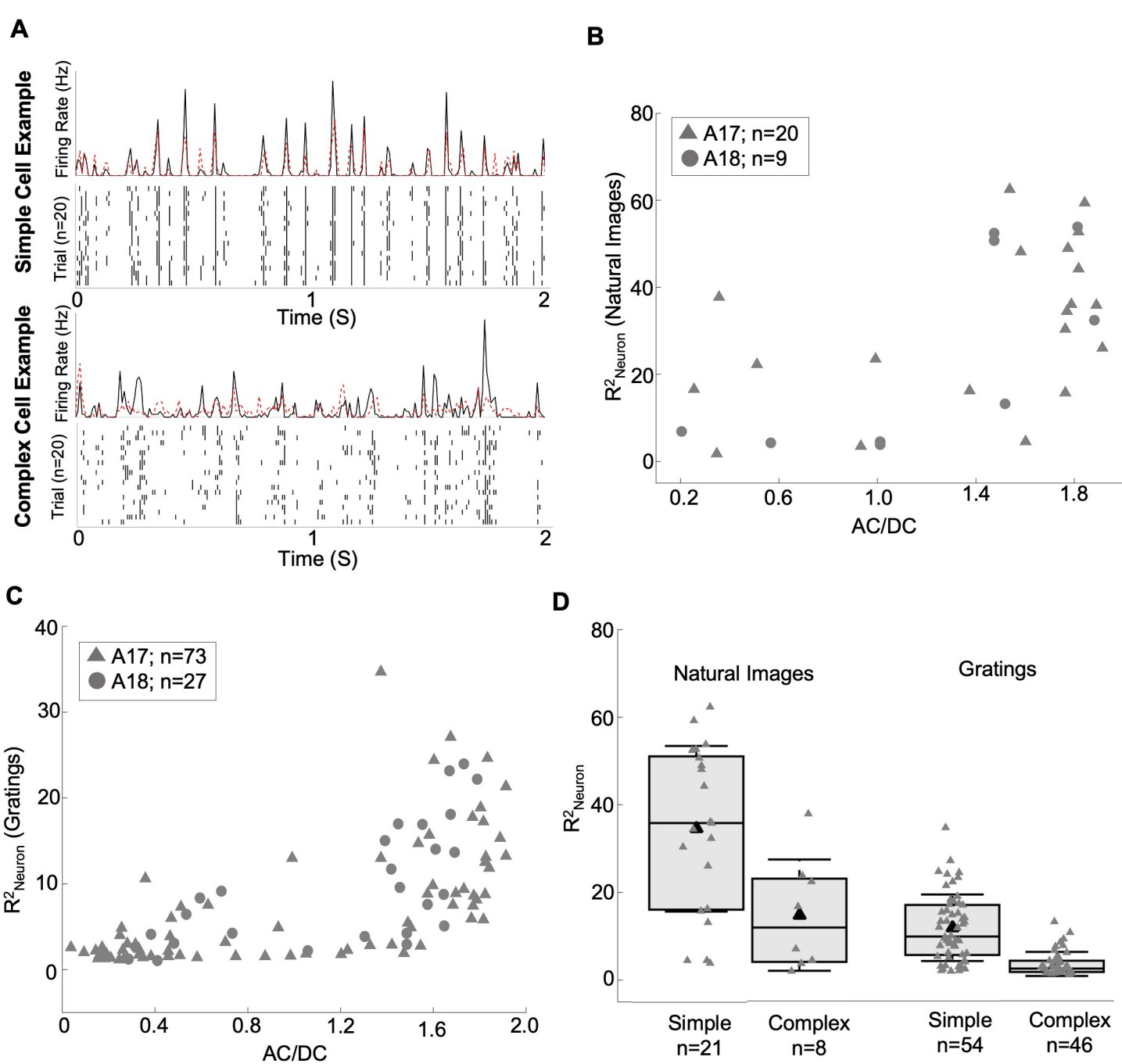

**Fig 7. Trial-by-trial reliability of neuronal responses for simple and complex type cells.** (A) Spike raster plots for responses of example simple and complex cells (same neurons as in Figs 3 and 4, respectively) to 20 repetitions of a natural image ensemble (only first 2 sec shown). Top traces are post stimulus time histograms (bins = 6.67 ms) for the neurons' spiking responses (black) and predicted responses of the convolutional model (red). (B) Neurons' reliability index, $R^2_{neuron}$, for natural image responses as a function of grating phase sensitivity, AC/DC (n = 29). Larger values of $R^2_{neuron}$ correspond to greater reliability. (C) $R^2_{neuron}$ calculated from responses to each neuron's preferred grating (n = 100). (D) Population averages of response reliability, $R^2_{neuron}$, of simple and complex type cells to gratings and to natural images. Conventions for box and whisker plots are same as in Fig 5A.

gratings, as a function of each neuron's AC/DC ratio. Again, it is clear that many of the simple cells show higher $R^2_{neuron}$ values than complex cells. For neurons classified as simple, the average $R^2_{neuron}$ was 7.6% (n = 54). For neurons classified as complex, the average $R^2_{neuron}$ was

3.3% (n = 46). The distribution for simple cells was significantly greater (t-test, p = 1.77e-10), again showing that complex cells were noisier and less reliable.

Thus the poorer predictive performance for complex cells (Fig 5) might be, at least in part, due to their greater trial-wise variability, instead of (or in addition to) an inadequacy of model architecture.

## Relationship between PReLU α and simple/complex type responses

In many situations, suitable grating data for obtaining AC/DC values for individual neurons may not be available. As explained earlier, the PReLU α parameter of the convolutional model distinguishes between a linear-nonlinear model and a hierarchical (subunit summation) model, which are generally thought to reasonably represent simple and complex cells, respectively. Thus, the PReLU α value of the convolutional model itself, as estimated from responses to broadband stimuli such as natural images, could provide a surrogate for the AC/DC value.

In order to examine the interpretability of the PReLU α, Fig 8A shows a comparison between the convolutional model's PReLU α obtained from natural image responses, and AC/DC values measured with grating stimuli for 29 neurons (20 from A17, 9 from A18). Neurons with an explainable VAF less than 1.0% were excluded, to help ensure reliability of the estimated model parameters. As expected, simple type cells (AC/DC > 1.0) indeed had higher PReLU α values, typically near unity, while complex type cells usually had values less than unity, sometimes less than zero. While the relationship was not strictly linear, the PReLU α and AC/DC were highly correlated, with a Pearson's correlation coefficient of r = 0.84 (p = 1.12e-08, n = 29). In terms of ranked values, these measures were also highly related, with a Spearman's correlation coefficient of r = 0.82 (p = 6.08e-08, n = 29). Thus, the PReLU α estimated from natural image responses could be used to distinguish simple vs. complex type cells with reasonable (if not perfect) reliability.

Since complex cells have a higher trial-to-trial response variability compared to simple cells, we can expect lower raw VAFs for complex cells with lower PReLU α values compared to simple cells with higher PReLU α values. Fig 8B shows this for the population of neurons, with lower predictive performance in terms of raw VAF for complex cells and higher predictive performance for simple cells. Across the population of neurons, we observed a significant correlation coefficient of r = 0.58 between raw VAF and the estimated PReLU α values (p = 3.49e-16, n = 168). In terms of ranked values, these measures were also highly related, with a Spearman's correlation coefficient of r = 0.59 (p = 0.0, n = 168).

The distributions of estimated alpha values for the full sample of neurons are shown as histograms in Fig 8C and 8D. The A17 sample (Fig 8C) appears to have a range of alpha values, with many neurons having alpha values close to unity, indicative of simple-type response properties, and others having values below zero, corresponding to complex-type responses. The distribution for the A18 cells (Fig 8D) shows a similar range of alpha values. However, neither of the distributions for both A17 and A18 cells indicate statistically significant bimodality by Hartigan's dip test (A17: p = 0.45, n = 104, dip at 0.03; A18: p = 0.83, n = 64, dip at 0.03).

We compared our convolutional model's predictive accuracy (VAF) to that of a version of the model with PReLU α fixed at 1.0 (equivalent to a simple LN model). The VAFs are comparable for simple cells with PReLU α close to 1.0, as expected (S3 Fig). However, for complex cells our model performed with a significantly higher VAF (S3 Fig, p = 2.3516e-05, n = 35).

## Spatial receptive field properties from the convolutional model

So far, we have emphasized the ability of the fitted CNN models to predict neurons' responses to novel visual stimuli which were not used for training—this is important for establishing the

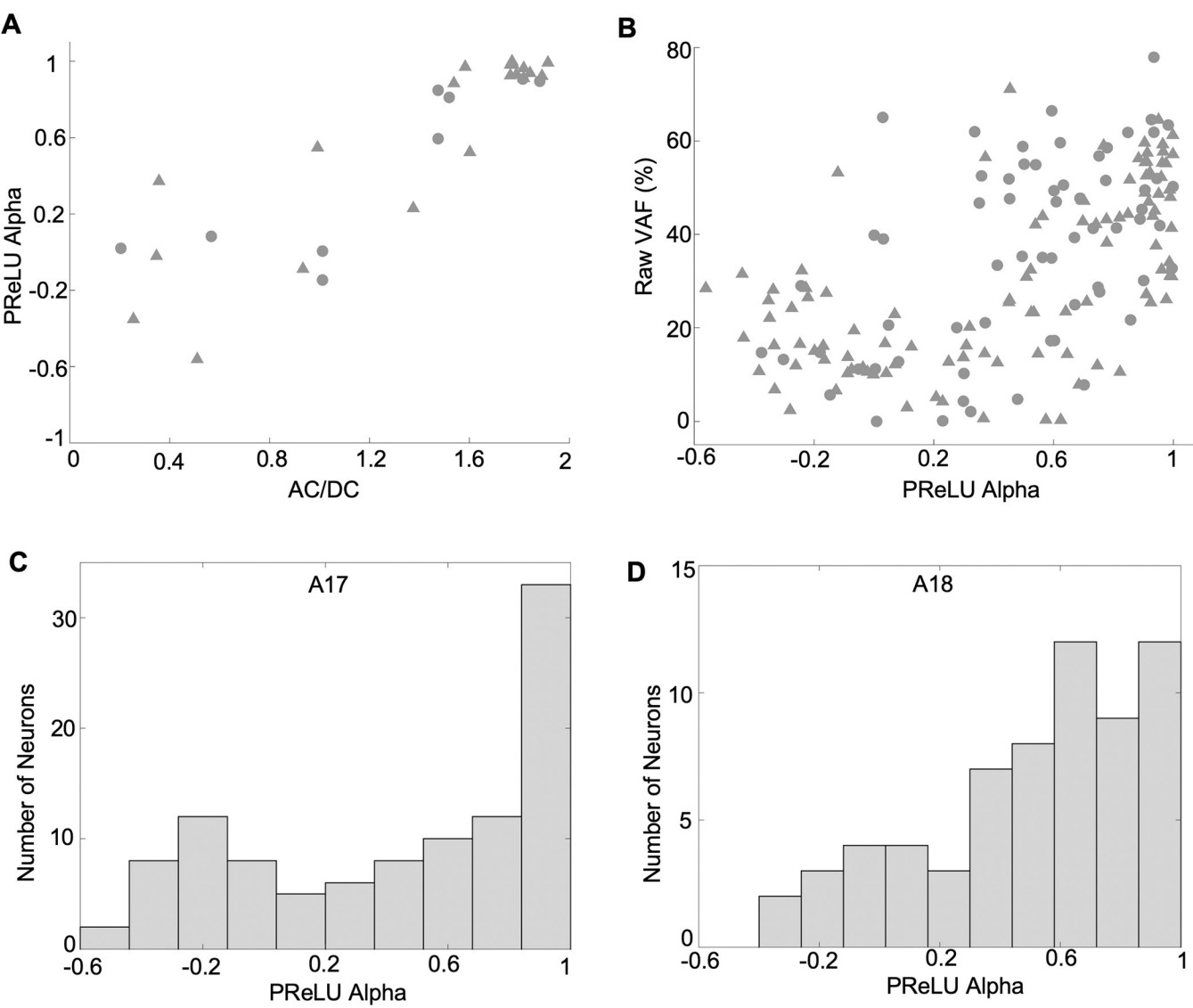

**Fig 8. Relationship between model prediction accuracy, simple-vs-complex type responses, and model PReLU.** (A) Comparison between the convolutional model's PReLU α parameter and neurons' phase sensitivity (AC/DC) (n = 29). (B) Comparison between model prediction accuracy and the convolutional model's PReLU α parameter (n = 168). (C) Histogram of alpha values for A17 neurons (n = 104) **D**, Histogram of alpha values for A18 neurons (n = 64).

predictive power of the fitted models for a wide variety of visual stimuli. However, an important utility of system identification is also to gain insight into each neuron's selectivity for particular properties of visual stimuli, and how these preferences vary from one neuron to another, and from one brain area to another.

As an illustration of how the CNN results can be used to characterize receptive fields, we fit Gabor functions to the reconstructed filter functions at the optimal temporal lag for each neuron (see Methods). To help avoid spurious function fits to noisy results, we included in this analysis only those neurons whose fitted CNN models predicted the test datasets with VAF values greater than 10% (94 out of 104 A17 cells, 57 out of 64 A18 cells). Two examples of such Gabor-fits are seen in Fig 9A, in each case showing the CNN reconstructed filter (left), best-fitting Gabor (middle), and residual (difference between them, right). A typical A17 cell with alternating, elongated excitatory and inhibitory regions is shown in the top row of Fig 9A—

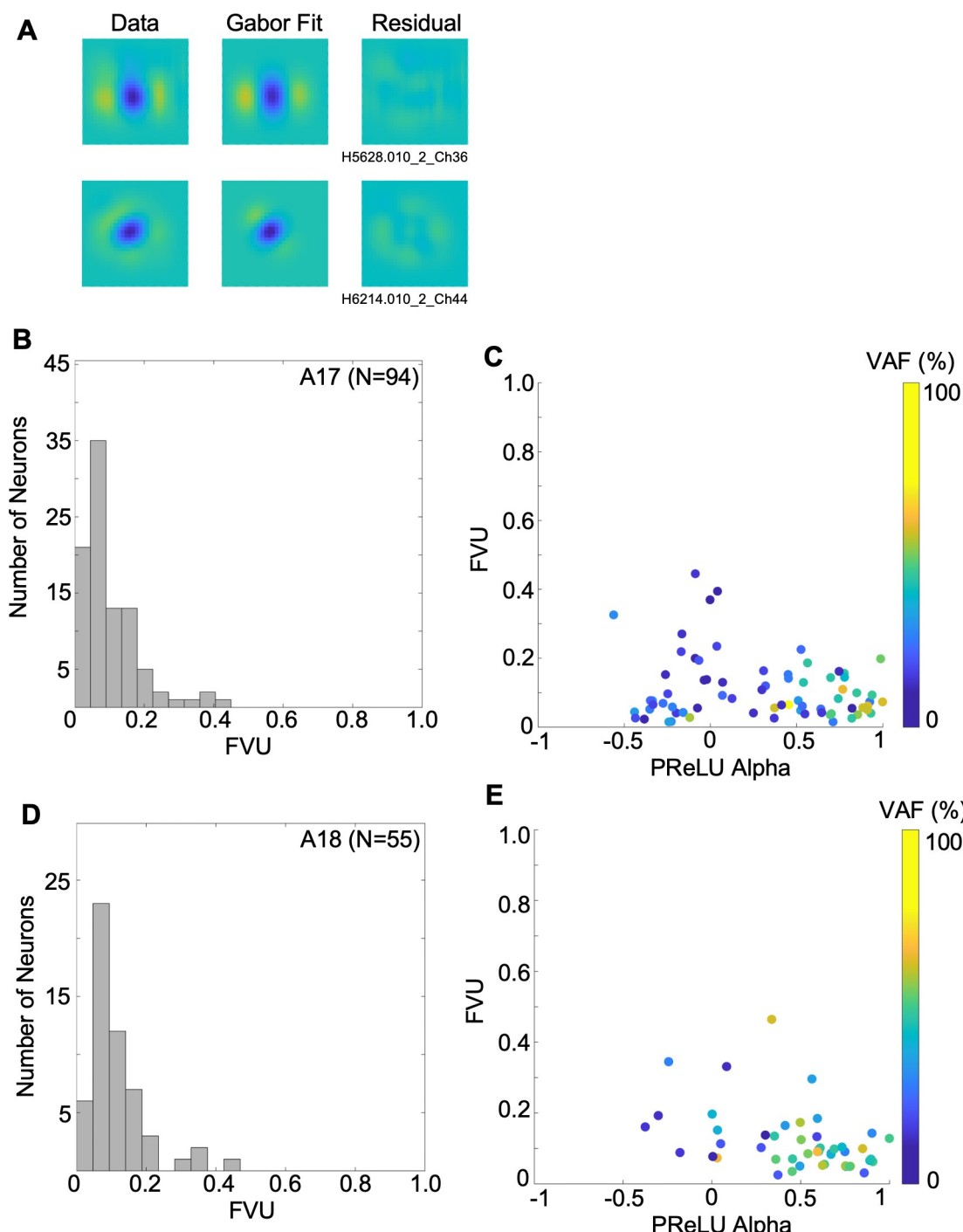

**Fig 9. Fitting of 2-D Gabor functions to receptive field restorations.** (A) 2-D Gabor fits to estimated spatial receptive field (convolution of filter with map layer) at optimal time lags, for same two example neurons as in Figs 3 and 4. Images in left column show the estimated receptive fields, with best-fitting Gabors in the middle column, and the residuals in the right column. (B) Histogram of FVU (fraction of variance unaccounted) values, for A17 sample. (C) scatterplot of FVU vs PReLU alpha values for A17 sample, with symbol for each neuron colored to indicate VAF for model's prediction of test dataset. (D) as in *B*, but for A18 sample. (E) as in *C*, but for A18 sample.

this was a complex-type cell (alpha = -0.12), giving a VAF = 15.1% and a very good Gabor function fit (fraction of unexplained variance, FVU = 0.04). The bottom row illustrates a simple-type cell (alpha = 0.8) with a relatively non-oriented, center-surround spatial receptive field, giving a VAF = 61.3% and a very good fit by the Gabor function (FVU = 0.09).

In agreement with previous studies using reverse correlation analyses of simple-type cells (e.g., [30,48,52]), most A17 receptive fields were fairly well fit by a Gabor function, as seen in the histogram of FVU values (Fig 9B)—the FVU values were small, ranging from 0.0148 to 0.445 (median = 0.0732). Such good fits were in spite of some neurons having receptive fields with very little orientation selectivity [30] or having "crescent" shaped flanking regions (e.g. Fig 3). These relatively good fits were found across the full range of PReLU $\alpha$ values (Fig 9C), showing that Gabor functions are also appropriate descriptions for the selectivity of complex as well as simple type cells. FVU was not significantly correlated with alpha in the A17 cells (r = -0.245, p = 0.018, n = 94). For A18 neurons, the FVU histogram in Fig 9D shows that Gabor functions gave similarly good fits in most cases, with FVU values from 0.0238 to 0.464 (median = 0.0906). We excluded from further analysis only two A18 neurons for giving a particularly poor fit (FVU > 0.5). For the remaining A18 neurons, the FVU was not significantly correlated with alpha, as seen in Fig 9E (r = -0.374, p = 0.048, n = 55).

The simplest receptive field parameter to examine is its size. Here we estimate RF size from the envelope of the best-fitting Gabor, as $(\sigma_{x'}^2 + \sigma_{y'}^2)^{0.5}$, where $\sigma_{x'}$ and $\sigma_{y'}$ are the envelope sigma parameters along the length and width, respectively. The estimated RF sizes for A17 neurons, illustrated in the histogram of Fig 10A, ranged from 0.538 to 10.05 deg (median 2.38 deg)—these values largely agree with previous results from cat A17 (e.g., [16,55]. A scatterplot of these RF sizes against the alpha values for the A17 sample is shown in Fig 10B, showing that simple and complex type cells had highly overlapping ranges of RF sizes, but with substantially larger values in a minority of complex-like cells. As a result, resultant correlation of RF size with alpha was statistically significant (r = -0.358, p = 4e-04, n = 94). A18 neurons had a wide range of RF sizes (1.40 to 13.1 deg, median = 5.18 deg), as seen in the histogram of Fig 10C. The ranges of sizes for lower vs. higher values of PReLU $\alpha$ were highly overlapping (Fig 10D), resulting in a non-significant correlation (r = -0.334, p = 0.012, n = 55). Comparison of the histograms in Fig 10A and 10C makes it clear that RF sizes tended to be larger in A18 than A17, confirmed statistically with an unequal variance (Welch) t-test (p = 9.4e-8).

A characteristic spatial property of early visual cortex neurons is their selectivity for spatial frequency. In our analysis a neuron's optimal spatial frequency would be captured explicitly by the frequency parameter ($f$) of the best-fitting Gabor function (Eq 5). Most neurons in A17 had optimal values ranging from 0.0049 to 0.689 cpd (Fig 11A), with a median of 0.187cpd (n = 94). This range is consistent with most previous reported measurements [55–57]. However, these frequencies are lower than those in some reports (e.g. [58]), probably due to the differing ranges of eccentricities of the receptive fields. Optimal spatial frequencies for lower vs. higher values of PReLU $\alpha$ (Fig 11B) overlapped greatly—this was confirmed by a non-significant correlation of optimal spatial frequencies with PReLU $\alpha$ (r = 0.2003, p = 0.0530, n = 94). Most neurons in A18 had estimated optimal spatial frequencies between 0.0088 and 0.2896 cpd (median 0.0646 cpd), as seen in the histogram of Fig 11C. Neurons with smaller, more complex-like PReLU $\alpha$ values and neurons with larger PReLU $\alpha$ values (simple-like cells) spanned overlapping range of frequencies (Fig 11D). This relationship did not show a significant correlation (r = 0.174, p = 0.205, n = 55). The $R^2_{Neuron}$ had no relation to the estimated tuning properties of neurons as shown by the non-significant correlation between optimal spatial frequency and $R^2_{Neuron}$ (S4 Fig, A17: r = 0.10, p = 0.30; A18: r = 0.01, p = 0.94). Comparison of the distributions for A17 vs A18 (Fig 11A and 11C) indicates a range of optimal spatial

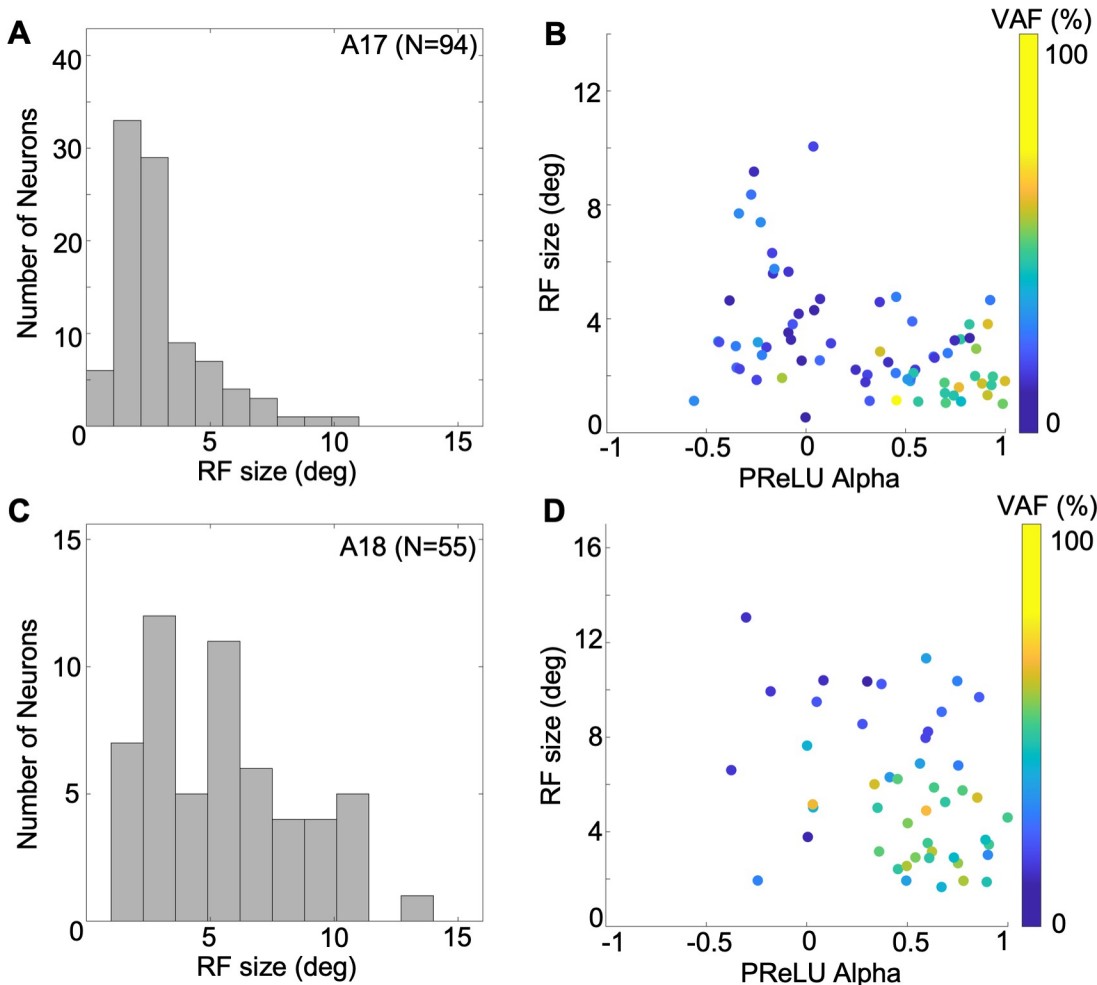

**Fig 10. Receptive field sizes estimated for simple and complex-type A17/18 cells, from envelopes of fitted Gabors.** (A) Histogram of estimated receptive field sizes for A17 cells (N = 94). (B) Relationship of A17 receptive field sizes to PReLU $\alpha$ values. Each plotted point represents one neuron, with colour indicating VAF. (C, D) same as A, B for A18 cells (N = 55).

frequencies of lower values in A18 than those in A17, consistent with previous findings for single units [56–58] as well as optical imaging [59,60] in the cat.

We noted that the estimated spatial frequencies were slightly higher when the Gabor functions were fitted to the filter weights instead of the restoration for both simple and complex cells in A17 and A18 (S5 Fig). This could be due to smoothing due to the convolution with the (Gaussian) map layer, which could shift the tuning to higher spatial frequency, thus biasing estimates of spatial frequency tuning to lower values. However, the quality of the Gabor fits was substantially lower when the filter weights were used, which might cause artifacts in estimated spatial frequencies (Figs 5A and 5C and S5). However for the neurons with acceptable FVU values, the estimated optimal spatial frequencies from the filter weights were highly correlated with (A17: r = 0.77, p = 2.5e-8; A18: r = 0.89, p = 4.4e-5), and only slightly lower (A17: 14.8%, A18: 29.8%), than those from the restorations (S5 Fig). Therefore in the above comparisons, Gabor fits on restorations were used in the above comparisons.

Spatial receptive fields of early cortical neurons also vary in their length (along the orientational axis), width, and ratio of length-to-width, or ratio [52]. An important refinement of this

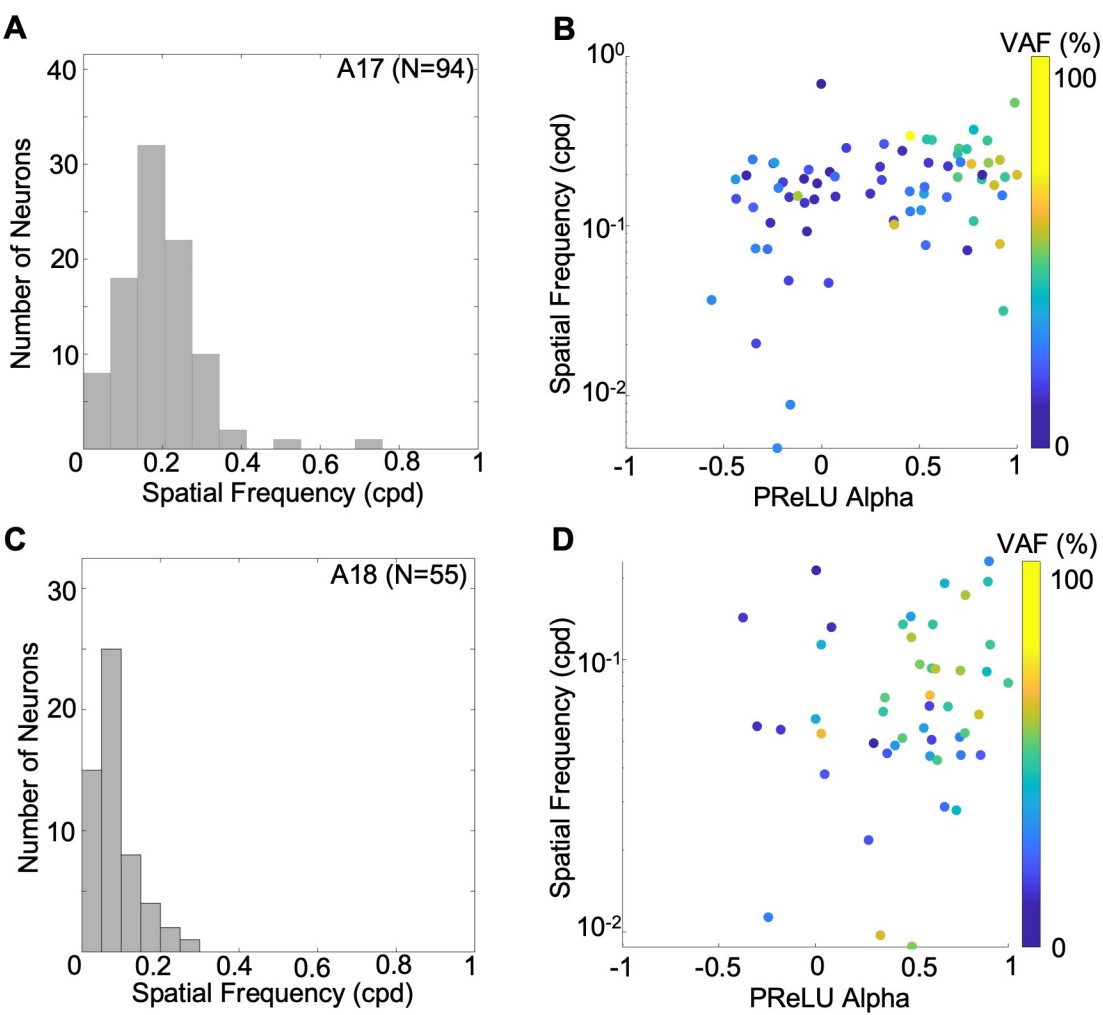

**Fig 11. Optimal spatial frequencies estimated for simple and complex-type A17/18 cells, from envelopes of fitted Gabors.**
(A) Histogram of estimated optimal spatial frequencies for A17 cells (N = 87). (B) Relationship of A17 optimal spatial frequencies to alpha values (Pearson's r = 0.200, p = 0.053). Each plotted point represents one neuron, with colour indicating VAF. (C, D) same as *A*, *B* for A18 cells (N = 49). (r = 0.173, p = 0.205).

analysis is to consider how receptive field length and width co-vary across the neuronal population. Here we employed the approach of Ringach [30], who considered width and length in terms of the number of cycles of the best-fit Gabor's optimal spatial frequency:

$$n_x = \sigma_x f \text{ and } n_y = \sigma_y f \tag{8}$$

where *f* is the spatial frequency and $\sigma_x$ and $\sigma_y$ are the envelope parameters along the width and length of the fitted 2d Gabor. Thus $n_x$ and $n_y$ are dimensionless and independent of other parameters from the Gabor fits—larger values indicate narrower tuning bandwidths for spatial frequency and orientation [30].

Fig 12A and 12B shows scatterplots of $n_y$ versus $n_x$ for our sample of A17 and A18 cells. In these plots, if the neurons' points fell on the dashed line (1:1 ratio), they would all have an aspect ratio of unity. However, in both cases note that many of the neurons' points do not lie on a straight line, indicating a wide range of aspect ratios. Furthermore, note that the smallest $n_x/n_y$ values (near the origin) fall close to the 1:1 line, while mid-range values mostly fall below

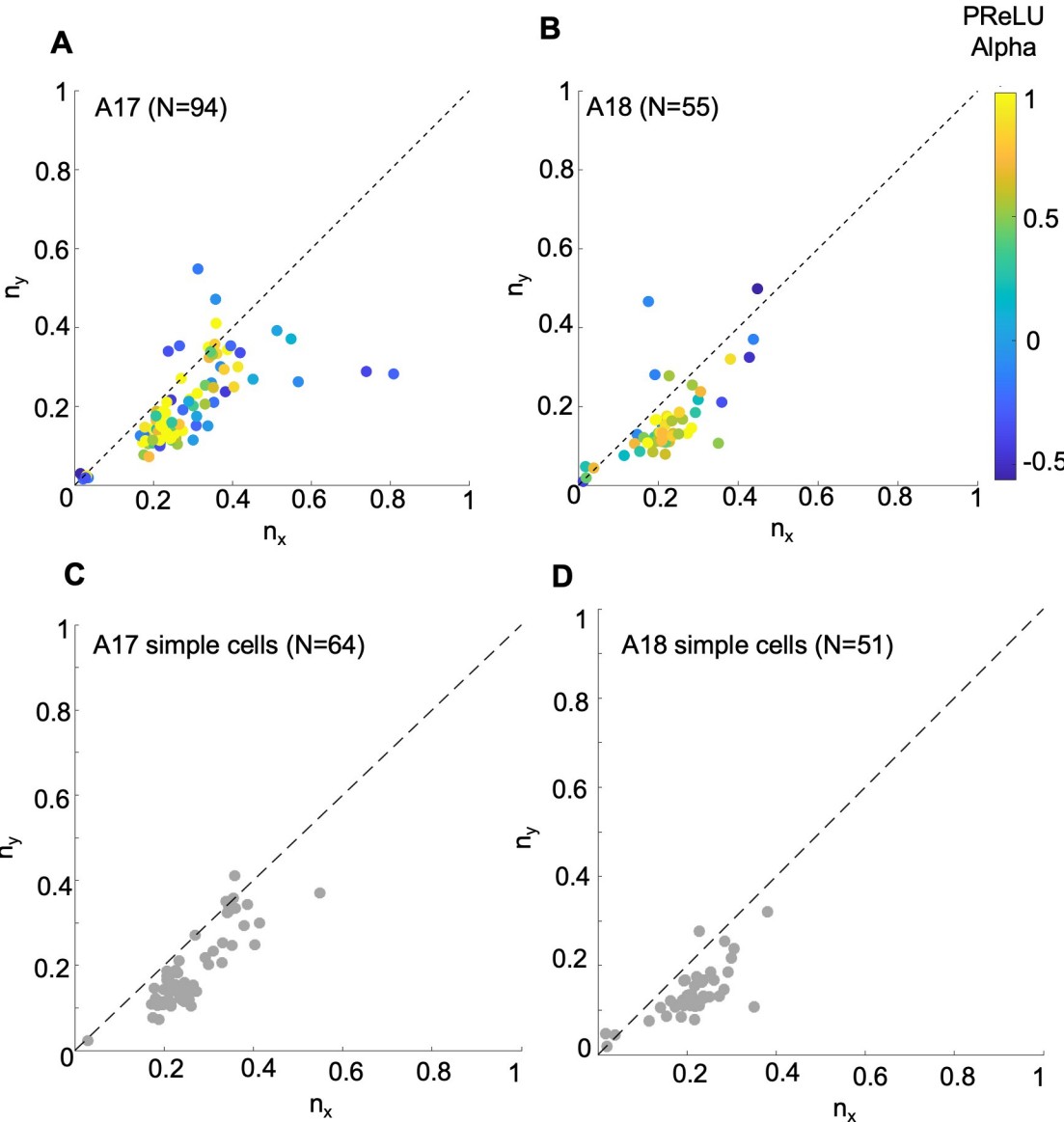

**Fig 12. Spatial bandwidth relationships for A17/18 cells, from envelopes of fitted Gabors.** Results for sampled neurons from (A) A17 (N = 94), (B) A18 (N = 55), (C) A17 simple cells with PReLU $\alpha$ >0 (N = 64), and (D) A18 simple cells with PReLU $\alpha$ >0 (N = 51). Each scatterplot shows relationship of $n_y$, the number of cycles along the length (optimal orientation), to $n_x$, the number of cycles along the width (orthogonal to optimal orientation), with colour indicating PReLU $\alpha$ values in A and B (simple-like cells more yellow, complex-like cells more blue). Dashed lines indicate 1:1 reference line.

the line, and larger values are more scattered. An earlier analysis using a reverse correlation method with a grating subspace, showed a similar kind of result for simple type cells in macaque V1, though with larger values occurring systematically above a 1:1 ratio [30]. The neurons' points in Fig 12A and 12B are colored to indicate the PReLU $\alpha$ values, suggesting that the neurons with higher $n_x/n_y$ values tended to have negative PReLU $\alpha$ values, indicative of complex-like cells that have higher response variability (Fig 7). Since the higher variability of these neurons might produce less reliable $n_x$ / $n_y$ values, and also to be comparable with earlier similar studies [30], we constructed the same scatterplots with these complex-like cells (PReLU $\alpha \leq 0$) removed. Results for the remaining population of more simple-like cells

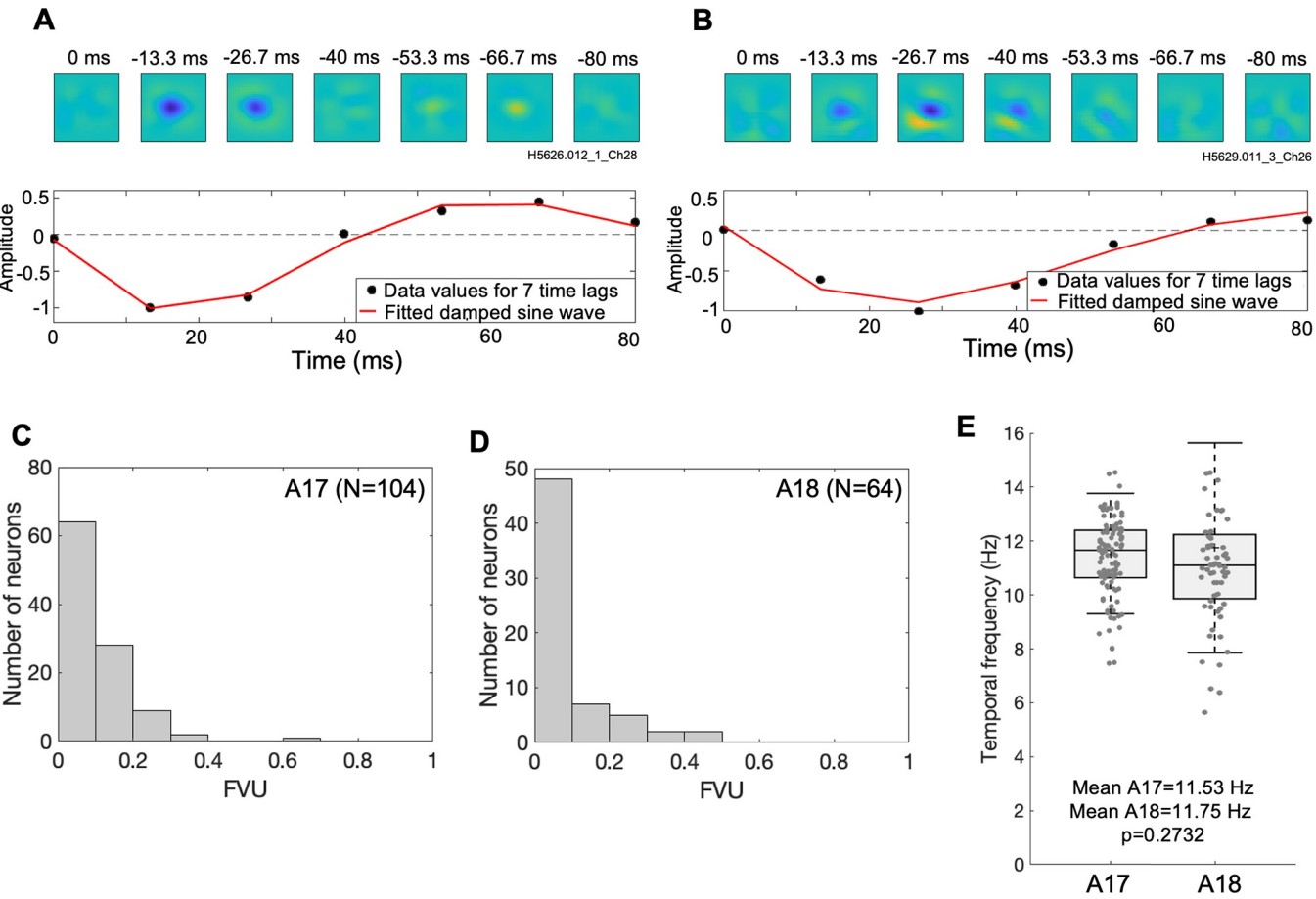

**Fig 13. Temporal dynamics of estimated RFs.** (A) Top panel: Linear restoration of STRF for an example neuron. Bottom panel: Estimated STRF amplitude (see text) tracked over 7 time lags in the STRF (black data points), and damped sinusoid curve-fit for these 7 time points (red trace). (B) Similar to A, for another example neuron. (C) Histogram of FVU (fraction of variance unaccounted) values from damped sinusoid fits, for A17 neuron. (D) as in C, but for A18 neuron. (E) Temporal frequencies of curve-fits for A17 neurons (mean f = 11.53Hz, n = 104) and A18 neurons (mean f = 11.75Hz, n = 64), respectively. Distributions indicated as box and whisker plots, where whiskers indicate standard deviations, box bounds indicate 25th and 75th percentiles, and horizontal lines indicate medians.

(PReLU $\alpha > 0$) are shown in Fig 12C and 12D for areas 17 and 18 respectively. These plots show a systematic relationship, more like that shown previously for macaque V1 simple cells [30], at least for the smaller and mid-range $n_x$ / $n_y$ values.

## Temporal receptive field properties from the convolutional model

An important benefit of estimating 3-dimensional spatiotemporal receptive fields is to gain insight into each neuron's temporal dynamics, and how they might vary across the population of neurons within and between different brain regions. Here we analysed the temporal dynamics of the estimated receptive fields by fitting a damped sinusoid function for selected pixel value changes across the seven measured time lags (see Methods).

Two examples of spatiotemporal receptive fields and corresponding temporal profiles are shown in Fig 13A and 13B. The example neuron in Fig 13A has an initial (negative) peak at 12 ms, goes through a polarity change at 40 ms and reaches a second (positive) peak at 68 ms. The strong biphasic temporal profile in Fig 13A is suggestive of a transient response to step-function stimuli, at least in a conventional linear model. Such polarity reversal is not evident in the

Fig 13B example neuron, containing only a primary peak at 28 ms, indicative of relatively more sustained responses (or possibly much slower biphasic profiles, not revealed by our limited range of temporal lags). Most A17 and A18 temporal dynamics, both biphasic and monophasic, were fairly well fit (red traces) by the early part of a damped sine wave function (Eq 7, Methods), as shown in histograms of FVU values (Fig 13C and 13D)—these values were small, ranging from 0.0075 to 0.648 (median = 0.0668) for A17 and from 0.0132 to 0.423 (median = 0.0514) in A18, indicative of relatively good fits.

The distributions of optimal temporal frequencies, inferred from the best-fitting damped sinusoids, are illustrated in Fig 13E. The A18 neurons had a slightly higher mean optimal temporal frequency across the population compared to those from A17 (A17: 11.53Hz vs A18: 11.75Hz). However the distributions for A17 and A18 were not significantly different (Mann-Whitney U-test, p = 0.2732).

## Discussion

We have shown how a simple convolutional model can estimate receptive fields of both simple- and complex-like visual cortex neurons from responses to natural images, and predict responses to holdback natural image ensembles as well as responses to sinewave gratings. We also demonstrate that the generally poorer predictive performance for complex cells is due in large part to a lower trial-wise reliability of responses from these neurons. In general, our spatial receptive field properties broadly agree with previous published reports, and extend earlier findings with laboratory stimuli (white noise, gratings) to responses to natural images, for both simple and complex cells in cat A17 and A18.

### PReLU $\alpha$ vs simple/complex-like nature of cortical neurons

The model's architectural simplicity (Fig 1) allows for a straightforward interpretation of its parameters. The filter layer can be thought of as a spatially delocalized simple cell, which primarily determines the neuron's selectivity for stimulus properties such as orientation and spatial frequency. The map layer indicates the region of space where the filter is active, i.e., the overall location of the receptive field. Most importantly, we find that a single parameter, the PReLU $\alpha$, can capture a neuron's spatial phase sensitivity. The PReLU $\alpha$ can be considered a rough representation of the balance between the filter and its opposite-polarity counterpart. When the two are balanced ($\alpha$ near 1.0), the PReLU approximates a linear relationship, so simple cell-like responses are generated. Not only does the PReLU $\alpha$ correlate well with the neuron's AC/DC value measured from grating responses, this parameter is also related to well-known models for early visual cortical neurons. The PReLU $\alpha$ values estimated from our sample of neurons varied in a continuum between -1 and +1, showing a range of simple- vs complex-like behavior of cortical neurons, but the distribution of PReLU $\alpha$ values did not show a clear bimodality. Thus comparing our results from natural image responses to those from gratings, these findings support the idea of a range of simple- and complex-like responses in early cortical neurons, but do not provide evidence for a categorical distinction.

One caveat of our study is that it is conceivable that the AC/DC ratio measured from grating responses might be biased to lower values (more complex-like) by greater noise. However, it seems unlikely that a similar effect would be evident for the PReLU alpha measured with our system identification.

As the PReLU $\alpha$ deviates from unity, the neurons' responses will begin to deviate from linearity. In this case, the model is more similar to the idea [61] that complex cell responses arise from the integration of multiple simple cells that act as nonlinear subunits. With complex cells, the interposed nonlinearity enables the possibility for system identification to retrieve the

subunit filters. However, with simple cells, due to the linearity, subunit filters cannot be explicitly recovered, since two linear filters in a cascade are equivalent to a single composite linear filter. The case where the PReLU $\alpha$ is near zero corresponds to Hubel and Wiesel's complex cell model, i.e., a linear summation of simple cells, all having the same filter properties but different positional offsets. Consider the filter layer, $w$, and a window of the stimulus, $s_1$. If the PReLU is a half-wave rectifier, $N(x)$, then $N(w \cdot s_1)$ acts like a linear-nonlinear model of a simple cell. The map layer indicates the region of visual space where the simple cell-like filter is active. If the PReLU $\alpha$ is negative, the model becomes somewhat more similar to an energy model [61] in which an opponent filter also contributes to the neuron's response.

However, this does not necessarily imply that the convolutional model corresponds to the underlying neural circuitry of visual cortex neurons—there are many different connectivity models that can generate simple- and complex-like responses. Rather, this convolutional model is a simple and reliable method of estimating the properties of early visual cortex cells. While the PReLU may show a simple relationship between simple and complex cells, its utility in the model does not provide concrete evidence for a particular underlying neural connectivity.

Although the convolutional model can perform well in predicting both simple and complex cell response properties, it does not provide a complete description of all early visual neurons. The intermediate nonlinearity is likely to be more complex than the PReLU, e.g. a polynomial function such as a square law, as in an energy model [61]. However, more elaborate functions generally increase the difficulty of the model parameter estimation. For example, a tent basis [38] could approximate a square law, however the required estimation procedure becomes more complicated. The simplicity of the PReLU allows for a smooth gradient calculation, and only adds one parameter to the model, which can be trained jointly with the other parameters. Another advantage of the PReLU is that its single parameter is simply related to the simple vs. complex cell distinction. Our method also does not require any additional initialization procedure; the weights can be initialized randomly (filter) or at a fixed value (PReLU/Gaussian).

It should be noted that despite the advantageous aspects of the convolutional model, the raw VAFs obtained for complex cells were lower than those for simple cells. However, these raw VAFs are in a similar range as those reported for other recently proposed estimation procedures [14,38]. We have shown that the complex cells are more noisy than simple cells in their trial-by-trial reliability. Using VAFs adjusted for this unreliability, we found that the relative difference between complex and simple cells was lowered, though not eliminated. This suggests that there is scope for improvement of our model architecture, to more fully explain complex cell responses.

## Aspect ratio of spatial receptive fields

Ringach et al. [30] used reverse-correlation to grating stimuli, to analyse the distribution of RF widths and lengths in terms of the number of cycles of the best-fit Gabor's optimal spatial frequency ($n_x$, $n_y$) for simple-type cells in macaque V1. He showed that the ($n_x$, $n_y$) distribution lies approximately along a one-dimensional curve, with non-oriented, broadband RFs near the origin and those with multiple elongated sub-fields (narrower bandwidths) farther from the origin. Using our convolutional model approach, we replicated that behaviour of neurons with lower $n_x/n_y$ values in both A17 and A18 of the cat. However our results for neurons with a higher $n_x/n_y$ appear quite scattered rather than falling along a one-dimensional curve—and we find few or no neurons with the largest magnitudes of $n_x$ or $n_y$ as observed in the macaque [30].

Conceivably this discrepancy might be due to the species difference—perhaps cat cortical neurons do not exhibit the narrow bandwidths for orientation and spatial frequency that can

be found in macaque V1. Another reason could be the different visual stimuli employed (i.e., natural images vs drifting sine wave gratings)—conceivably, the most narrowband-tuned neurons might respond strongly to narrowband grating stimuli but poorly to broadband natural image stimuli, and thus be less represented in our sample.

Also the results reported by Jones and Palmer [52] based on Gabor fits to cat A17 receptive fields estimated by reverse correlation, as replotted in [30], also showed neurons' $n_x/n_y$ values approximately along a one-dimensional locus, though mostly falling above the unity line on this kind of plot. This difference from our results as well as the macaque results of [30], could conceivably be due to the stimuli employed—i.e., the 2D sparse white noise of [52].

In our view the more important commonality in all these results, is the implication of a constraint on possible shapes of receptive fields, such that they lie along a one-dimensional continuum in the $n_x/n_y$ space. Filters produced by unsupervised learning methods applied to natural images [30,62] also indicate similar one-dimensional loci, suggesting that this constraint is related to efficient coding of natural scenes.

## Temporal dynamics

Previous studies with sine wave grating stimuli showed that A17 neurons respond well to lower temporal frequencies in the range of 2–4 Hz, and that A18 neurons respond better for comparatively higher temporal frequencies between 2–8 Hz [58,63,64]. These brain area-specific temporal frequency differences could be due to the predominance of Y cell input to A18 and X cell input to A17 from the LGN [65]. Our results showed optimal temporal frequency tuning up to 15Hz for A17 and A18, with A18 neurons exhibiting slightly higher optimal temporal frequencies compared to A17—but this difference did not show clear statistical significance (Fig 13E, p = 0.2732).

These discrepancies in estimated temporal properties could be due to the differences in visual stimuli employed—i.e. drifting sine wave gratings in previous studies compared to natural images here. For example, temporal frequency tuning can differ for stimuli with natural and whitened temporal dynamics, possibly reflecting adaptation to stimuli that do not change rapidly [17]. The two kinds of stimuli that we have used differ substantially in temporal correlation (much greater for drifting gratings), and in temporal frequency bandwidth (much greater for our natural images, which were presented abruptly and in rapid succession). However, it seems unclear how these differences might underlie the higher optimal temporal frequencies, or the similar temporal frequency tuning in A17 vs A18. Other previous studies have demonstrated substantially different temporal dynamics when estimated with different kinds of temporal presentation [66,67]. Our stimulus presentation is admittedly not very "natural", e.g. the temporal frequency spectrum is not fractal [68], however, we present a new image every 13.3 ms, which is qualitatively similar to saccade and microsaccade eye movements in natural vision (though the temporal rate here is more rapid than with saccades). Reid et al [66] showed that a sum-of-sinusoids approach to system identification revealed faster temporal properties of the recorded neural responses than expected from single sinusoid responses. Also Tolhurst et al [67] showed that A17 responses to abruptly presented sinewave gratings were much more transient than would be predicted from their temporal frequency dependence for sine wave gratings. Therefore, our temporally broadband, abrupt presentations of natural images might be causing the observed higher estimated optimal temporal frequencies in both A17 and A18, compared to grating responses. These differences presumably arise from some kind of nonlinearity in the temporal dynamics, such as contrast gain control [67,69], which is not incorporated in our model architecture.

## Comparison to previous system identification approaches

Vintch et al. [38] proposed a convolutional model for V1 receptive fields, having two filters capable of characterizing both simple and complex cells' responses to sparse white noise. However, estimation of the model parameters apparently required carefully chosen initial conditions and iterative back-and-forth estimation of different parts of the model. McFarland et al [39, 70] proposed a "nonlinear input model (NIM)" comprised of multiple linear-nonlinear (LN) units, to identify multiple stimulus features driving the neuron, followed by a common spiking nonlinearity. Unlike our model, this method estimates different numbers of LN subunits depending on the simple or complex-like nature of the modelled neuron. For some complex-type cells, a model architecture with multiple filters might be more appropriate compared to our model. However both the latter approaches would be at the expense of many additional model parameters. More recent approaches [19,25–28] employed deep neural network models, which can be quite complex and challenging to interpret. In all these approaches, the model parameters are not straightforwardly related to the simple/complex distinction. Due to the additional model parameters, such models would probably provide somewhat better predictive performance than the one described here, but for the same reason they may be more difficult to estimate. A future possibility might be to use the approach described here to obtain good initial conditions for a more complicated model such as the aforementioned ones.

A slight complication in estimating the convolutional model is the need to optimize the hyperparameter—in our model, a filter size should be tuned for each neuron. Fortunately, nearby neurons in the visual cortex typically have fairly similar sizes and/or preferred spatial frequencies [60], so the best model filters of neighboring neurons are likely to be similar in size. Thus, a hyperparameter search may not be necessary for every neuron. However in some cases, a limited hyperparameter search may be valuable, though the model can still perform fairly well with sub-optimal filter size.

An important advantage of the system identification approach used here is that it does not make an *apriori* assumption of a functional filter form, e.g., 2-D Gabor function, but rather it estimates all points in the filter from the neuron's responses. Consequently, this allows the method to reveal other kinds of receptive field shapes (several example RFs are shown in S6 Fig), including many that are non-oriented, or "crescent" shapes [71]. Then subsequently, we can examine how well candidate functions can serve as descriptions of the receptive field (e.g., Fig 9).

Since our convolutional model has only a single filter, it cannot capture properties such as cross-orientation inhibition [72], which requires at least two filters. Other properties such as surround suppression, contrast gain control [73], or second-order responses [74], also would require more complex model architectures with additional filters. However with improvements such as greater amounts of data, or additional methods of regularization [75], it may become possible to resolve greater model structure. Because the model is constructed within the framework of deep neural networks, with widely used tools for machine learning, further improvements to it can benefit from on-going rapid advances in software algorithms and GPU parallel processing support for this kind of learning algorithm.

Our system identification model architecture should be applicable to map receptive fields at early stages of visual processing in any species, provided they do not have multiple types of subunits, or require models with more linear-nonlinear stages (i.e. deeper neural network models). For example, our approach should be able to capture receptive fields, including ones that are not "Gabor-like", measured by reverse-correlation or by spike-triggered covariance, e.g. Fig 11 in Niell & Stryker [76].

An important finding with recordings from awake, behaving mice, has been that V1 neurons can respond not only to visual stimuli, but also exhibit responses related to locomotion

[76] or other spontaneous behaviors [77]. Our model obviously does not incorporate such motor-related responses, but it could be extended with the addition of regressors against signals related to such behaviours.

## Comparison to deep neural network approaches

Neurons in higher-tier visual areas require models with more filters and multiple layers [78,79]. For example, neurons in higher stages of visual processing become more selective for particular types of stimuli, but less positionally selective within larger receptive fields. A well-known example are neurons selective for faces, regardless of position or rotation. Models for these higher regions generally involve a hierarchy of processing stages (layers) beyond early-stage simple and complex cells, to build more complex and object-specific responses [78]. In a convolutional neural network model, this is achieved by adding more convolutional layers. Interestingly, it has been found that the activity of intermediate-layer neurons in large multi-layered convolutional neural networks have similarities to ventral stream cell responses [79].

Our convolutional model architecture has relatively few additional parameters beyond those of the linear-nonlinear model, making it straightforward to estimate. The PReLU imposes negligible computational cost—it has a single additional parameter that can be estimated within the gradient descent framework. Another important advantage of this simplicity is that the estimation is relatively robust with choice of initial conditions—therefore pre-training methods are unnecessary. The estimation of the convolutional model is sufficiently fast that it could conceivably be employed on-line during recording experiments (for one or a few neurons) to provide a rapid estimate of neuronal filtering and receptive field locations. In any case, the approach could be particularly valuable in the context of multi-neuron recording with multielectrode or 2-photon imaging, when different neurons may have quite different stimulus selectivities and receptive field locations.

## Supporting information

**S1 Fig. Example neurons requiring 1,2, and 3 passes for cropping.** First 3 example neurons require 3 passes and the receptive fields across 3 passes are shown. Second set of three example neurons require only 2 passes and the last set of 3 neurons require only 1 pass. (TIF)

**S2 Fig. Prediction of grating tuning curves by estimated models.** Spatial frequency tuning curve (average firing rate, normalized to maximum) for 6 example A17 neurons. Temporal frequency, 2 Hz; average of 10 repetitions of 1 sec each. Solid black line for neuron's response to the grating stimuli, red for tuning curve predicted by convolutional model. (TIF)

**S3 Fig. Model comparison to a simple LN model.** Comparison between predictive performance (test VAF) for the convolutional model and a simpler linear-nonlinear (LN) model. Each point denotes a neuron, color coded according to its estimated PReLU $\alpha$ parameter. (TIF)

**S4 Fig. Spatial frequency vs reliability ratio.** A) scatterplot of optimal spatial frequencies vs reliability ratio ($R^2$) values for neurons in the A17 sample. B) Similar to A, for A18 neurons. (TIF)

**S5 Fig. Comparison of Gabor fitting to filter weights vs linear restoration.** A) Histogram of FVU (fraction of variance unaccounted) values, for A17 sample when Gabor functions were fit to the filter (grey) and restoration (blue). B) Scatter plot of optimal spatial frequency estimated

when Gabor was fit to filter vs restoration—each point represents one neuron, color coded according to its PReLU alpha value. C) Similar to A for A18 neurons. D) Similar to B for A18 neurons.
(TIF)

**S6 Fig. Example receptive fields for 20 neurons, illustrating a variety of shapes.**
(TIF)

## Acknowledgments

We thank Guangxing Li for contributions to software and technical support in experiments; Amol Gharat for important contributions to data collection and spike-sorting; and Thomas Naselaris, for suggesting the parametrized Gaussian map layer in system identification.

## Author Contributions

**Conceptualization:** Philippe Nguyen, Jinani Sooriyaarachchi, Curtis L. Baker, Jr.

**Data curation:** Philippe Nguyen, Jinani Sooriyaarachchi.

**Formal analysis:** Philippe Nguyen, Jinani Sooriyaarachchi.

**Funding acquisition:** Curtis L. Baker, Jr.

**Investigation:** Philippe Nguyen, Jinani Sooriyaarachchi, Curtis L. Baker, Jr.

**Methodology:** Jinani Sooriyaarachchi, Qianyu Huang, Curtis L. Baker, Jr.

**Project administration:** Curtis L. Baker, Jr.

**Software:** Philippe Nguyen, Jinani Sooriyaarachchi, Qianyu Huang, Curtis L. Baker, Jr.

**Supervision:** Curtis L. Baker, Jr.

**Validation:** Philippe Nguyen, Jinani Sooriyaarachchi.

**Visualization:** Philippe Nguyen, Jinani Sooriyaarachchi, Qianyu Huang.

**Writing – original draft:** Philippe Nguyen, Jinani Sooriyaarachchi.

**Writing – review & editing:** Jinani Sooriyaarachchi, Curtis L. Baker, Jr.

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
