## [Decision Letter · Decision Letter 0]

18 Jan 2024

Dear Dr Baker Jr,

Thank you very much for submitting your manuscript "Estimating receptive fields of simple and complex cells in early visual cortex: A convolutional neural network model with parameterized rectification" for consideration at PLOS Computational Biology.

As with all papers reviewed by the journal, your manuscript was reviewed by members of the editorial board and by several independent reviewers. In light of the reviews (below this email), we would like to invite the resubmission of a significantly-revised version that takes into account the reviewers' comments.

Dear Curtis,

As you will read in the detailed reviews, the referees found your synthesis of ML and more classical methods for the estimation of RF quite interesting BUT they also had some major concerns that need to be addressed. I noticed for example that all three reviewers make a comment on the initial choice (for the size or cropping) of the filter layer. I believe you will be able to address most of these issues and will look forward to your detailed reply.

Frederic Theunissen

We cannot make any decision about publication until we have seen the revised manuscript and your response to the reviewers' comments. Your revised manuscript is also likely to be sent to reviewers for further evaluation.

Sincerely,

Frédéric E. Theunissen

Academic Editor

PLOS Computational Biology

Daniele Marinazzo

Section Editor

PLOS Computational Biology

Dear Curtis,

As you will read in the detailed reviews, the referees found your synthesis of ML and more classical methods for the estimation of RF quite interesting BUT they also had some major concerns that need to be addressed. I noticed for example that all three reviewers make a comment on the initial choice (for the size or cropping) of the filter layer. I believe you will be able to address most of these issues and will look forward to your detailed reply.

Frederic Theunissen

Reviewer's Responses to Questions

**Comments to the Authors:**

Reviewer #1: This manuscript develops a new model of early cortical visual processing that blends empirically-derived and machine-learning based analytical approaches. The authors analyze spiking data from cat A17 and A18 recorded during presentation of a large natural image sequence. Following a linear convolution layer, the model imposes a specialized nonlinear activation function designed to capture properties of simple and complex cells, a spatial summation and a final rectifying nonlinearity. The authors evaluate model performance and show that it can be used to measure several aspects of visual tuning, typically measured with synthetic stimuli.

The authors make a good case for their approach, which uses elements of machine learning to introduce flexibility to model fits without the explosion of parameters required for more generalized DNNs. Overall, the model and fit procedure are described quite clearly. The manuscript is also thorough in discussing the relationship of the current study’s findings to previous characterizations of tuning and encoding model analysis. There are some concerns. In particular, it is not clear if/how the use of a “reconstruction” to characterize spatial tuning interacts with the intervening nonlinearity. It also would be helpful if the authors could provide a quantitative comparison of model performance with one or more previously published models.

MAJOR CONCERNS

1. The use of “reconstruction” – convolving the summation layer with the convolutional layer – to smooth and visualize tuning is interesting. In the case of a purely simple cell (alpha=1), it makes sense (linear+linear = linear). However, for a complex cell, say with alpha=0, it seems like the convolution would have the effect of smoothing out high spatial frequency tuning that might be present in the initial convolutional layer, thus biasing estimates of spatial frequency tuning to lower values. Perhaps this concern was addressed in the manuscript, so it’d be great if the authors could clarify if this is/is not a concern. Alternatively, is it possible to fit the Gabor to the convolutional layer alone (without “reconstruction,” for some neurons with high SNR, perhaps) to test for possible bias in measurements of tuning for complex cells.

Lesser, but related, the term “reconstruction” is potentially confusing to readers. Previously this term has been used to describe decoding analysis where stimuli are reconstructed from neural population activity. The authors might consider a different term.

2. Is there a concern that variations in explainable variance (R^2_neuron) could impact measurements of tuning properties? For example, if a unit is overall noisier, will the AC/DC ratio trend toward lower values? It’s great that the authors consider noise ceilings in their analysis, but it seems important that they also determine if any differences in tuning (e.g., complex vs simple cells, A17 vs. A18) is robust to differences in response reliability. One idea for addressing this would be to degrade high-R^2_neuron spike trains to values that match noisier neurons and show that the reported differences are preserved when response SNR is matched. There are likely to be alternative controls that accomplish the same thing, but the authors should consider how response reliability can impact measures of tuning.

3. The model appears to perform reasonably well, in terms of variance explained. However, someone considering use of this approach would want to know if it is really more effective than alternative, perhaps more established encoding models. It may not be practical to perform an exhaustive comparison of all possible alternatives. But is it possible to fit a traditional LN model and compare performance? A very quick and dirty LN model would be to freeze alpha at 1 and fit the other parameters. Or, is there published data that has been fit with a different model that could be run through the current code? Some kind of quantitative comparison would be informative.

LESSER

p. 2 “compensation for this unreliability” This phrasing is confusing. The term “noise ceiling” is reasonably well established and might be useful to frame this result. (as in the author summary)

p. 6-7. The authors might note early work by Prenger at al. (Neural Networks 2004) that fit full spatio-temporal models using ANNs.

p. 11. Can the authors report the number of distinct images that were presented in each subset (training/regularization/testing). 20 x 375 = 7500 unique images. Is this correct? Were the stimuli and groupings identical for all neurons?

p. 11-14. It’s helpful that the authors identify free parameters and point out their relatively low number compared, e.g., to CNN models. Can they report the range of total free parameters required for the complete model? It does appear to vary depending on the size of the convolutional filter. Is that the only source of variation?

p. 16. Was any regularization imposed on the model fits? An L2 penalty for the convolutional layer would probably not improve performance substantially, but it might clean up the fits.

p. 17. R^2_neuron. Is there a reference for this approach to measuring the noise ceiling? It appears that he 19-rep PSTH is treated as a sort of ground truth response against which a noisy response is compared, but can’t the 19-rep PSTH still be noisy if the unit has a very unreliable response. Maybe the logic just needs to be spelled out more clearly.

p. 19. “time lag of maximal variance” Were the linear filters typically space-time separable? If so, one could possibly improve spatial tuning estimates by computing the first principal component of the filter in space vs. time.

p. 23. Fig. 2C. Can the authors superimpose the individual data points in this box plot? Same request for some of the later box plots.

It would also be helpful if the authors could explain what “linear reconstruction” (or some alternative term, see above) means when they plot it in Fig 2.

p. 27. Fig. 5A. Can the authors provide a similar plot showing noise-corrected model performance?

p. 35. “FVU” is this 1-VAF/100? Is there a reason for switching to a different error metric here?

p. 44. The use of PReLU is a nice innovation. It is slightly confusing, though, in if/how it would deal with a cell that is, say the sum of three simple cells in quadrature phase (ie, a sum of simple cells that is not purely complex). Is this a limitation of the PReLU approach vs. a subunit model?

p. 48. The authors might note that David et al 2004 specifically observed that temporal frequency tuning differed between stimulus with natural and whitened temporal dynamics, possibly reflecting adaptation to stimuli that do not change rapidly.

TYPO/GRAMMAR

p. 4 “However, predicting responses … through system identification [3].” There’s something funny about this sentence.

p. 27. Fig. 5 legend. “holdback”. Should it be “held-out”?

p. 44. “it’s opposite” should be “its opposite”

Reviewer #2: Nguyen et al. used a convolutional model with parameterized rectification to estimate receptive field properties of simple and complex cells in cat areas 17 and 18. They found that the convolutional model could predict the responses of simple cells better than those of complex cells to both natural images and drifting gratings. The poor predictive performance of complex cells was primarily due to low trial-by-trial reliability. Moreover, they found that the parameterized rectified linear unit (PReLU) value correlated well with the AC/DC ratio, a traditional measurement for simple/complex segregation. However, the distribution of the PReLU values was not bimodal, indicating that simple and complex are not discrete cell types. The authors also showed that the receptive field properties measured with natural images are comparable to those with artificial images (such as sparse noise and drifting gratings) in cat areas 17 and 18.

Overall, the authors provide a straightforward model that can capture many features of simple and complex cells. The methods are described in detail, and the discussion appropriately covers most of the findings. However, a couple of issues need to be sufficiently addressed.

Major:

1. Some important terms and claims need to be addressed appropriately. Please provide additional statements and explanations for the following points.

-- The authors did not specify how the filter layer and its size were determined. The method used to define the filter layer is critical because it determines the features of simple and complex cells, including the number of subunits, preferred orientation, spatial frequency, etc. What’s the procedure for the optimization? This question is critical because the choice of window size largely affects the fitting wellness and the VAF values.

-- The authors claimed that the relative difference in the VAF between complex and simple cells was lower by eliminating the trail-by-trail unreliability of complex cells. Please provide data to support the claim.

-- The trial-by-trail reliability for simple and complex cells is much higher for natural image responses than drifting gratings responses. Please provide explanations for the difference.

2. There are many discrepancies in receptive field properties, such as preferred spatial frequency and aspect ratio (nx and ny), between measurements in the study and those in the literature. The contradicting results are probably due to the difference in stimuli used for receptive field mapping: the authors used natural images, whereas other pieces of literature used sparse noise. The authors should recheck the data, explain these discrepancies in the discussion, and include more cat literature on spatial features. Here are the details.

-- Fig.11A. The preferred spatial frequency in most area 17 neurons is less than 0.4 cpd. In contrast, Movshon et al. (1978) reported a 0.3-3.0 cpd range.

-- Fig. 12. Most of the data points in the distribution of nx and ny are below the unity (1:1 ratio) line, but Jones and Palmer (1987, studying cat simple-cell receptive fields, Table 1, replotted in Ringach 2002) reported the opposite – most of the data points are above the unity line.

Minor:

1. Page 10. Drifting gratings were used to measure the AC/DC ratio and spatial frequency tuning. Please specify the optimized grating (i.e., size, contrast, orientation, temporal frequency).

2. Page 22. The authors used “R2neuron < 1% and explainable VAF < 1%” to exclude neurons with insufficient response reliability. What’s the rationale behind these criteria because one percent seems to be very low?

3. Page 23-26. In Figs. 2-4, the window size of the linear reconstruction is twice as large as that of the filter. If the purpose of these figures is to represent the similarity of the maps increase with the cropping procedure, it would be better if the two window sizes were the same. Furthermore, the scale bar is marked with only “+” and “-“. Is the map normalized by dividing the absolute maximum value of the map?

4. Page 28. In Fig. 6, I suggest the authors add a third figure representing the comparison between simple and complex cells (just like Fig. 5B) because that’s one of the main themes in the manuscript.

5. Page 47. The discussion regarding the aspect ratio of spatial receptive fields should include the literature on cat areas 17 and 18. Receptive field properties in cat and monkey V1 are similar but not identical. For example, the receptive field size is smaller, and the preferred spatial frequency is higher in monkey V1 than cat V1.

Reviewer #3: In this manuscript, the authors develop a simple convolutional neural network model approach to estimate spatio-temporal receptive fields in A17/18 of the anesthetized cat by predicting responses to natural images. The most interesting aspect of the model in my view is that its parameters can be related to biological properties of visual cortical neurons in relatively straightforward ways. In addition, the model is able to recover both simple and complex cell RFs, and predicts a conti

---

## [Decision Letter · Decision Letter 1]

1 May 2024

Dear Dr Baker Jr,

We are pleased to inform you that your manuscript 'Estimating receptive fields of simple and complex cells in early visual cortex: A convolutional neural network model with parameterized rectification' has been provisionally accepted for publication in PLOS Computational Biology.

Best regards,

Frédéric E. Theunissen

Academic Editor

PLOS Computational Biology

Daniele Marinazzo

Section Editor

PLOS Computational Biology

Thank you for addressing all the comments. Please see the minor revisions requested by Reviewer 3.

Best wishes,

Frederic Theunissen.

Reviewer's Responses to Questions

**Comments to the Authors:**

Reviewer #1: The authors have done a good job addressing the issues raised during review, clarifying how their results should be interpreted and the relation of this study to previous work.

Reviewer #2: The authors have appropriately addressed all of my comments. I don't have any more questions.

Reviewer #3: I would like to thank the authors for carefully considering my comments, for their replies and for incorporating the changes into the manuscript. I also would like to apologise for the confusion regarding the model species in the Cadena et al study that I had mentioned in my assessment, which was monkey and not mice, as pointed out by the authors.

In my view, the revision has further improved the quality and clarity of the manuscript. I only have a few minor points of clarification left:

- Figure 1 and legend: for clarity, could you please try to indicate time also on the left side of the figure? Also, I find the sentence “Cyan elements represent estimable parameters.” somewhat confusing, because there seems to be free parameters that appear in other colours, e.g. green / red. Likewise, I only see cyan in the surround of the filter layer and the Gaussian Map layer. Please also explain for the last panel the choice of colours (i.e. red = predicted?)

-Figure 3: seems to be the same neuron as in Figure 2. If correct, please indicate this in the figure legend

- p. 27: “However, the difference between simple vs. complex mean exp VAFs (13.9%) is relatively lower (22.4%) compared to the difference between raw VAFs.” To enhance clarity, please consider moving the “(22.4%)” to after “the difference between raw VAFs”. Also, please support the statement with a statistical test.

- Description of Fig. 7D, please add the results of the supporting statistical tests (“The R2neuron is higher for responses to natural images compared to gratings (Fig 7D)”).

**Have the authors made all data and (if applicable) computational code underlying the findings in their manuscript fully available?**

Reviewer #1: Yes

Reviewer #2: Yes

Reviewer #3: Yes

PLOS authors have the option to publish the peer review history of their article (what does this mean?). If published, this will include your full peer review and any attached files.

Reviewer #1: No

Reviewer #2: No

Reviewer #3: **Yes: **Laura Busse

---

## [Editor Report · Acceptance letter]

28 May 2024

PCOMPBIOL-D-23-02009R1 

Estimating receptive fields of simple and complex cells in early visual cortex: A convolutional neural network model with parameterized rectification

Dear Dr Baker Jr,

I am pleased to inform you that your manuscript has been formally accepted for publication in PLOS Computational Biology. Your manuscript is now with our production department and you will be notified of the publication date in due course.

With kind regards,

Olena Szabo
